# CALIBRATED UNCERTAINTY SAMPLING FOR ACTIVE LEARNING

## ABSTRACT

We study the problem of actively learning a classifier with a low calibration error. One of the most popular Acquisition Functions (AFs) in pool-based Active Learning (AL) is querying by the model's uncertainty. However, we recognize that an uncalibrated uncertainty model on the unlabeled pool may significantly affect the AF effectiveness, leading to sub-optimal generalization and high calibration error on unseen data. Deep Neural Networks (DNNs) make it even worse as the model uncertainty from DNN is usually uncalibrated. Therefore, we propose a new AF by estimating calibration errors and query samples with the highest calibration error before leveraging DNN uncertainty. Specifically, we utilize a kernel calibration error estimator under the covariate shift and formally show that AL with this AF eventually leads to a bounded calibration error on the unlabeled pool and unseen test data. Empirically, our proposed method surpasses other AF baselines by having a lower calibration and generalization error across pool-based AL settings.

## 1 INTRODUCTION

Active Learning (AL) has recently become a crucial topic in machine learning due to the development of Deep Neural Network (DNN) and Big data. In the pool-based sequential AL, we consider situations where the unlabeled data pool is abundant but manual labeling is expensive (e.g., medical diagnosis, etc.). Given a fixed budget labeling cost per round, our goal is to design an Acquisition Function (AF) to actively query informative samples from the expert for labels, such that the model can quickly learn and generalize well on unseen data (Roy & McCallum, 2001; Settles, 2009).

To achieve this goal, uncertainty-based approaches (e.g., least-confident, entropy (Shannon, 1948; Wang & Shang, 2014), margin sampling (Roth & Small, 2006), BALD (Gal et al., 2017)) and diversity-based approaches (e.g., Coreset (Sener & Savarese, 2018), BADGE (Ash et al., 2020)) have shown promising results by achieving a better generalization than the naive random-sampling baseline (Lüth et al., 2023; Citovsky et al., 2021; Kim et al., 2020). However, such approaches often focus solely on generalization (e.g., accuracy) and overlook the verification of model uncertainty estimation quality (e.g., calibration) (Tran et al., 2022; Nalisnick et al., 2019).

Meanwhile, the ability of models to produce high-quality uncertainty estimation is crucial for a reliable DNN in high-stakes applications (e.g., forecasting, healthcare, finance, etc.). Beyond the generalization, in principle, a reliable model also includes uncertainty estimation ability to permit graceful failure, signaling when it is likely to be wrong (Liu et al., 2020; Bui & Liu, 2024). That said, the uncertainty estimation quality of the model, especially derived from DNN, is often poorly calibrated (Kuleshov et al., 2018; Nado et al., 2021). Therefore, this results in two critical issues in the literature on AL (e.g., Fig. 1). Firstly, the uncertainty quantification quality of the aforementioned AL baselines on unseen test data is not verified, leading to high risks in high-stakes applications. Secondly, suppose the model is uncalibrated on the unlabeled pool during the query times, then the AF with uncertainty-based sampling is also unreliable, resulting in non-informative query samples and leading to sub-optimal generalization and high calibration error on unseen data. This is because the uncertainty sampling method in AL follows the intuition that AF should be "querying the most uncertain samples in the unlabeled pool because they may be the most inaccurate". However, if the model is uncalibrated, then the most uncertain samples may not be the most inaccurate, resulting in inefficient uncertainty sampling AF.

Figure 1: Accuracy and calibration comparison on MNIST with the number of labeling rounds $T = 50$ and labeling cost $k = 10$. Uncalibrated-least-conf is the least-conf method (i.e., the least confident based on the softmax probability, details in Apd. B.1), but is additionally made to be uncalibrated by randomly scaling the logit vectors for every sample. We can see that when the model is uncalibrated, the least-confident sampling not only has a worse Expected Calibration Error (ECE) than our method but also has a lower accuracy because of querying non-informative samples. *A short demo is available at this Google Colab link.*

Therefore, with the goal of developing a reliable model in this pool-based AL setting, we propose a new AF strategy in Fig. 2 by ranking based on the lexicographic order of the calibration error and the model uncertainty. Specifically, we estimate the calibration error of each sample on the unlabeled pool by using the kernel trick with the cumulative labeled data. Then, using the lexicographic ordering of Cartesian products (Duffus, 2006; Murphy, 2023), we select samples with the highest calibration errors. If samples share the same calibration error, we continue to select samples with the highest model uncertainty. The key ideas of this approach are: (1) selecting the samples with the highest calibration error can help the model with self-calibrated predictive certainty, maintaining a high uncertainty estimation quality on the unlabeled pool and unseen dataset; (2) using this well-calibrated model can help uncertainty sampling discover informative samples during query times, improving calibration and generalization.

Our contributions can be summarized as follows:

1. We propose **C**alibrated **U**ncertainty **S**ampling for AL (**CUSAL**), an AF by estimating calibration error on the unlabeled pool and using the lexicographic order of calibration and uncertainty, prioritizing querying samples with the highest calibration error before leveraging model uncertainty.
2. We theoretically provide an upper bound for our calibration estimator error under the covariate shift of AL. Furthermore, we prove that using our AF, we can bound the expected calibration error of the learned classifier on the unlabeled pool and the unseen test data.
3. We empirically confirm our proposed method surpasses other acquisition strategies in the pool-based AL by having a lower calibration and generalization error across several settings on the MNIST, F-MNIST, SVHN, CIFAR-10, CIFAR-10-LT, and ImageNet datasets.

## 2 PRELIMINARY

### 2.1 PROBLEM SETTING

We denote $\mathcal{X}$ and $\mathcal{Y}$ as the sample and label space. A dataset is defined by a joint distribution $\mathbb{P}(X, Y) \in \mathcal{P}_{\mathcal{X} \times \mathcal{Y}}$, where $\mathcal{P}_{\mathcal{X} \times \mathcal{Y}}$ is the set of joint probability distributions on $\mathcal{X} \times \mathcal{Y}$. We consider the pool-based AL setup (Roy & McCallum, 2001), i.e., we are given a warm-up dataset $S_0 = \{(x_i, y_i)\}_{i=1}^{n_0}$, where $n_0$ is the number of data points in $S_0$ and $(x_i, y_i) \sim \mathbb{P}(X, Y)$, $\forall i \in [n_0]$. We are also given an unlabeled dataset $U_0 = \{x_j\}_{j=n_0+1}^{n_0+m_0}$, where $m_0$ is the number of data points in $U_0$, s.t., $x_j \sim \mathbb{P}(X)$ and can request the true corresponding label sample $y_j$ according to $\mathbb{P}(Y|X)$ for every $j \in n_0 + [m_0]$. We aim to test a learning model on the test set $D_{te} = \{(x_i, y_i)\}_{i=1}^{n_{te}}$, where $n_{te}$ is the number of data points in $D_{te}$, s.t., $(x_i, y_i) \sim \mathbb{P}(X, Y)$, $\forall i \in [n_{te}]$.

We can consider the standard pool-based AL as a sequential decision-making process, i.e., at round $t \in [T]$, where $T$ is the number of labeling rounds, the model agent parameterized by a learnable parameter $\theta_t$, is provided a sequence of unlabeled pool $U_t \subseteq \mathcal{X}$ and seeks to find a global minimum

$$x_t^* := \arg\max_{x \in U_t} AF(x; \theta_{t-1}), \tag{1}$$

where $AF$ is an Acquisition Function. Once the set of points $\{x_t^*\}$ has been chosen, the oracle $\mathbb{P}(Y|X)$ provides its corresponding label $\{y_t^*\}$. Hence, the new-labeled set and the pool set become

$$S_{t+1} = S_t \cup \{x_t^*, y_t^*\}, \quad U_{t+1} = U_t \setminus \{x_t^*\}. \tag{2}$$

And, the model parameter $\theta_t$ is optimized (i.e., trained) with the labeled set $S_{t+1}$, after which it receives a loss outcome $\ell_{te}(\theta_t)$ on the test set $D_{te}$, and repeat this process at the next round $t + 1$.

## 2.2 GENERALIZATION

Under the classification setting, the model predicts $y \in \mathcal{Y}$, where $\mathcal{Y}$ is discrete with $K$ possible categories by using a forecast $h(\cdot) := h(\cdot; \theta) = \sigma \circ f(\cdot; \theta)$, which composites a backbone $f(\cdot; \theta) : \mathcal{X} \to \mathbb{R}^K$, parameterized by $\theta$, and a softmax layer $\sigma : \mathbb{R}^K \to \Delta_y$ which outputs a probability distribution $W(y) : \mathcal{Y} \to [0, 1]$ within the set $\Delta_y$ of distributions over $\mathcal{Y}$; the value of probability density function of $W$ is $w$. Similar to the literature on AL, we consider $\ell_{te}(\theta_t)$ to include the generalization error by the accuracy of $h$, i.e.,

$$\mathbb{E}_{(x,y)\sim\mathbb{P}(X,Y)}\left[\ell_{01}(h(x), y)\right] := \mathbb{E}_{(x,y)\sim\mathbb{P}(X,Y)}\left[\mathbb{I}[\arg\max_{y_k \in \mathcal{Y}} [h(x)]_{y_k} \neq y]\right]. \quad (3)$$

## 2.3 CALIBRATION

Furthermore, we consider the calibration error in Eq. 4 as the second term of $\ell_{te}(\theta_t)$. Specifically, let us first recall the definition of distribution calibration (Dawid, 1982) for the forecast:

**Definition 2.1.** (Song et al., 2019; Kuleshov et al., 2018) A forecast $h$ is said to be **distributionally calibrated** if

$$\mathbb{P}(Y = y|h(x) = W) = w(y), \forall y \in \mathcal{Y}, W \in \Delta_y.$$

Intuitively, the forecast $h$ is well-calibrated if its outputs truthfully quantify the predictive uncertainty. For instance, if we take all data points $x$ for which the model predicts $[h(x)]_y = 0.8$, we expect $80\%$ of them to take on the label $y$ indeed. This yields the definition of the $L_p$ calibration error of $h$ as follows

$$CE_p(h) := \left(\mathbb{E}\left[\|\mathbb{E}[Y|h(X)] - h(X)\|_p^p\right]\right)^{\frac{1}{p}}, \quad (4)$$

where

$$\mathbb{E}[Y|h(X) = h(x)] = \frac{\sum_{y_k \in \mathcal{Y}} y_k p(h(X) = h(x), Y = y_k)}{p(h(X) = h(x))}. \quad (5)$$

Since the distributionally calibrated Def. 2.1, we can say $h$ is perfectly calibrated if $CE_p(h)^p = 0$.

# 3 CALIBRATED UNCERTAINTY SAMPLING METHOD

To improve calibration and generalization performance, we introduce our novel AF by ranking based on the lexicographic order of the calibration error and the model uncertainty. We formally present our framework as follows:

## 3.1 ESTIMATING CALIBRATION ERROR ON UNLABELED POOL

To select the highest calibration error, we need to know the calibration error of each sample on the unlabeled pool. Unfortunately, as provided in Eq. 4, this quantity requires the knowledge of the true label $y$, and this is an unknown quantity for our model in the AL setting.

Therefore, to estimate the calibration error of each sample on the unlabeled pool, firstly, let us recall the consistent and differentiable Lp canonical calibration error estimator for $n$ Independent-identically-distributed (IID) labeled samples $\{(x_i, y_i)\}_{i=1}^n$ from the joint distribution $\mathbb{P}(X, Y)$ (Popordanoska et al., 2022). In particular, let us consider the estimator of canonical calibration error, where the expected value outside in Eq. 4 is measured by the empirical data, i.e.,

$$\widehat{CE_p(h)}^p := \frac{1}{n}\sum_{j=1}^n \widehat{CE_p(h(x_j))}^p = \frac{1}{n}\sum_{j=1}^n \left\|\mathbb{E}_{iid}\left[\widehat{Y|h(x_j)}\right] - h(x_j)\right\|_p^p, \quad (6)$$

where

$$\mathbb{E}_{iid}\left[\widehat{Y|h(x_j)}\right] := \frac{\sum_{i=1}^n k\left(h(x_j); h(x_i)\right) y_i}{\sum_{i=1}^n k\left(h(x_j); h(x_i)\right)}, \forall j \in [n]. \quad (7)$$

As verified by Popordanoska et al. (2022), when $p(h(x))$ is Lipschitz continuous over the interior of the simplex, $\exists$ a kernel s.t. $\mathbb{E}_{iid}\left[\widehat{Y|h(x_j)}\right]$ is a pointwise consistent estimator of $\mathbb{E}[Y|h(x_j)]$, that is:

$$\plim_{n\to\infty} \frac{\sum_{i=1}^n k\left(h(x_j); h(x_i)\right) y_i}{\sum_{i=1}^n k\left(h(x_j); h(x_i)\right)} = \mathbb{E}[Y|h(x_j)]. \quad (8)$$

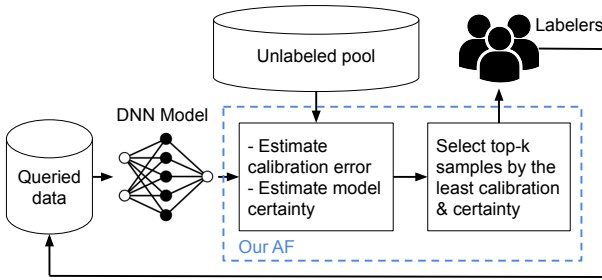

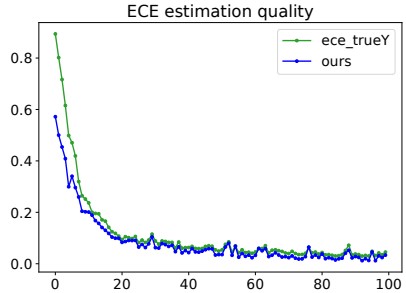

Figure 2: Overview of our calibrated uncertainty sampling framework for Active Learning.

Figure 3: ECE estimation quality between known labels in Eq. 7 and unknown labels (ours) in Eq. 9 on MNIST in Sec. 5.3.

From Eq. 8, firstly, we can see that to estimate calibration error $\mathbb{E}\left[Y|h(x_j)\right]$ for every sample $j \in [n]$, it requires the knowledge of $n$ labels. This requirement is not met on the unlabeled pool of AL setting. Secondly, it is worth noticing that to verify the consistency, i.e., Eq. 8, Popordanoska et al. (2022) requires $n$ samples are sampled IID. That said, this assumption does not hold in AL setting by the covariate shift, i.e., the difference in distribution between unlabeled pool samples $U$ and the queried samples $S$ from the model $h$ (Liu et al., 2015; Prinster et al., 2024). So, the challenge here is how we can modify the estimator in Eq. 6 such that it is still consistent on the unlabeled pool.

To address this challenge in AL setting, we modify the estimator in Eq. 6 to estimate the calibration error quantity on the unlabeled pool. Specifically, for every round $t \in [T]$, given $n_t$ samples in the new-labeled set $S_t = \{(x_i, y_i)\}_{i=1}^{n_t}$ and $m_t$ samples in the unlabeled pool set $U_t = \{x_j\}_{j=n_t+1}^{n_t+m_t}$, our new estimator for every sample on the unlabeled pool $x_j$, is as follows:

$$\widehat{CE_p(h(x_j))}^p := \left\| \frac{\sum_{i=1}^{n_t} k\left(h(x_j); h(x_i)\right) y_i}{\sum_{i=1}^{n_t} k\left(h(x_j); h(x_i)\right)} - h(x_j) \right\|_p^p, \forall j \in n_t + [m_t], \tag{9}$$

where $k$ is a Dirichlet kernel, i.e.,

$$k\left(h(x_j); h(x_i)\right) := \frac{\Gamma(\sum_{k=1}^{K} \alpha_{ik})}{\prod_{k=1}^{K} \Gamma(\alpha_{ik})} \prod_{k=1}^{K} h(x_j)_k^{\alpha_{ik}-1}, \tag{10}$$

with $\alpha_i = \frac{h(x_i)}{b} + 1$, where $b$ is a bandwidth parameter in the kernel density estimate (experiments with other kernels, including RBF-Gaussian, cosine, and linear kernel, are in Fig. 19). It is worth noting that while the estimator in Eq. 6 & Eq. 7 requires both $i, j$ in the same set $[n]$, our estimator in Eq. 9 estimate $j$ in $m_t$ unlabeled sample $U_t$ from $i$ in $n_t$ queried datapoints $S_t$. These $S_t$ and $U_t$ are two separate sets. Hence, we formally prove our calibration estimator in Eq. 9 is also point-wise consistent and provides its estimator error bound under the covariate shift of AL setting in Thm 4.1. Intuitively, if the model knows the calibration error of every sample on the unlabeled pool, it can query the highest calibration error samples to self-correct their confidence prediction in the future. Following this idea, we propose our new AF in the next section.

### 3.2 OUR LEXICOGRAPHIC ORDER ACQUISITION FUNCTION

Following our calibration error estimator in Eq. 9, we leverage it for our novel AF as follows. Our key idea is that if the model is unreliable, we should fix its calibration error, i.e., choose the least calibrated samples first, and then use uncertainty sampling later to query informative samples in the unlabeled pool. Formally, we consider the two partially ordered sets on the unlabeled pool, including the calibrated error set $A$ and the uncertainty set $B$ from the softmax layer, defined as follows

$$A := \left\{ \left\| \frac{\sum_{i=1}^{n_t} k\left(h(x); h(x_i)\right) y_i}{\sum_{i=1}^{n_t} k\left(h(x); h(x_i)\right)} - h(x) \right\|_p^p \mid x \in U_t \right\}, \quad B := \left\{ \max_{y \in [K]} [h(x)]_y \mid x \in U_t \right\}. \tag{11}$$

Then, our AF is

$$x_t^* = \arg\max_{x \in U_t} AF(x; \theta_{t-1}), \tag{12}$$

where $x$ is chosen by the lexicographical order $(a, b)$ on the Cartesian product $A \times B$, which is defined as

$$(a, b) \leq (a', b') \Leftrightarrow a < a' \text{ or } (a = a' \text{ and } b \geq b'), \tag{13}$$

---

**Algorithm 1** Our AF: **C**alibrated **U**ncertainty **S**ampling for AL (**CUSAL**) (code is in Apd. B.2)

---

**Input:** Labeling cost $k$, labeling rounds $T$, warm-up and queried dataset $S_0$, unlabel-pool set $U_0$, test set $D_{te}$, and model $h$.
Train $h$ on $S_0$.
**for** $t = 0 \rightarrow T$ **do**
    Estimate calibration error of $h$ on every sample in $U_t$ by Eq. 9 with $S_t$.
    Select top-$k$ samples $\{x_t^*\}$ on $U_t$ by the lexicographic order in Eq. 12.
    Receive the corresponding labels $\{y_t^*\}$ according to $\mathbb{P}(Y|X)$ for the set of points $\{x_t^*\}$.
    $S_{t+1} \leftarrow S_t \cup \{x_t^*, y_t^*\}$.
    $U_{t+1} \leftarrow U_t \setminus \{x_t^*\}$.
    Train $h$ on $S_{t+1}$.
    Test accuracy and calibration of $h$ on $D_{te}$.
**end for**

---

for all $a, a' \in A$ and $b, b' \in B$. Since we denote $A$ as the set of estimated calibration errors and $B$ as the set of model uncertainties for samples on the unlabeled pool $U_t$, the AF in Eq. 12 means we select samples with the highest calibration errors. If samples share the same calibration error, we continue to select samples with the highest model uncertainty. The pseudo-code for our algorithm is presented in Alg. 1. Intuitively, the acquisition strategy in Eq. 12 suggests that when $h$ is uncalibrated, we will select the least calibrated samples in the unlabeled pool to improve uncertainty quantification quality. And, when $h$ becomes reliable by being perfectly calibrated (i.e., the calibration error across samples is the same and small), the AF will select the least confident samples to improve generalization performance. We formally explain this behavior in the following section.

## 4 THEORETICAL ANALYSIS OF OUR ACQUISITION FUNCTION

Firstly, recall that in Eq. 8, Popordanoska et al. (2022) shows that their estimator $\mathbb{E}_{iid}\left[\widehat{Y|h(x_j)}\right]$, $\forall j \in n$ in Eq. 7 is a pointwise consistent estimator of $\mathbb{E}\left[Y|h(x_j)\right]$. Yet, this result only holds when the $n$ data points are sampled IID, i.e., $\{x_i, y_i\}_{i \in [n]} \sim \mathbb{P}(X, Y)$. Unfortunately, this does not hold in AL. Specifically, this is due to the biased sampling during query times from the AF of the model leading to a covariate shift existing between the cumulative labeled data $S$ and samples from the unlabeled pool $U$ (Liu et al., 2015; Dasgupta, 2011; Fannjiang et al., 2022). Formally:

$$S \sim \tilde{\mathbb{P}}(X)\mathbb{P}(Y|X), \quad U \sim \tilde{\mathbb{P}}(X_S)\mathbb{P}(Y|X), \tag{14}$$

where $\tilde{\mathbb{P}}(X)$ denotes the marginal distribution of samples selected from the model $h$, $\tilde{\mathbb{P}}(X_S)$ denotes the marginal distribution of $X$ excluding the samples random variable $X_S$ in the cumulative labeled dataset $S$. It is worth noticing that at the start of AL, although all data points, i.e., $S_0 \cup U_0$ could be sampled IID from $\mathbb{P}(X, Y)$, the covariate shift happens when we update our model on $S_0$, then we use this trained model to query in the next round. For example, if we were trying to predict the next president based on voting during a presidential election in the U.S., although the votes of people are sampled from the same distribution of U.S. citizens, after querying for the most informative samples, the model could end up biased to a sub-population distribution (e.g., citizens in N.Y. state), resulting in a covariate shift between the cumulative and unlabeled pool.

Therefore, we next guarantee that our modified estimator in Eq. 9 is a point-wise consistent estimator under the covariate shift of AL by the following theorem:

**Theorem 4.1.** *Given a sample $x$ on the unlabeled pool with $m_t$ data points and $n_t$ samples in the cumulative labeled data, our estimator in Eq. 9 is a point-wise consistent estimator under active learning with covariate shift, i.e.,*

$$\operatorname*{plim}_{n_t \to \infty} \left\{ \mathbb{E}_{\pi_U}\left[\widehat{Y|h(x)}\right] := \frac{\sum_{i=1}^{n_t} k\left(h(x); h(x_i)\right) y_i}{\sum_{i=1}^{n_t} k\left(h(x); h(x_i)\right)} \right\} = \mathbb{E}_{\pi_U}[Y|h(x)],$$

*where $\pi_U$ denotes the sample distribution on the unlabeled pool. And, the mean square error of our estimator in Eq. 7 is bounded by*

$$\mathbb{E}\left[\left|\mathbb{E}_{\pi_U}\left[\widehat{Y|h(x)}\right] - \mathbb{E}_{\pi_U}[Y|h(x)]\right|^2\right] \leq \mathcal{O}(n_t^{-1} b^{\frac{-K+1}{2}} + b^2).$$

The proof is in Apd. A.1 with the assumption that $n_t$ samples are sampled i.i.d. w.r.t. $\mathbb{P}_s(X,Y) = \tilde{\mathbb{P}}(X)\mathbb{P}(Y|X)$. This assumption based on the fact that for every round $t \in [T]$, the queried samples $\{(x_i, y_i)\}_{i=n_{t-1}+1}^{n_t}$ are sampled IID w.r.t. $\mathbb{P}_s(X,Y)$, when $h(x)$ is a probabilistic acquisition function. This is also discussed and mentioned in the covariate shift in AL (Liu et al., 2015; Fannjiang et al., 2022). From Thm. 4.1, we can see that when $n_t$, i.e., the number of samples in the warm-up data or cumulative pool, is large enough, we can achieve a perfect calibration error estimator. In addition, when $n_t$ increases, the quality of the estimator in Eq. 9 also improves correspondingly. We verify this behavior in Fig. 3 with a decrease in the gap between our estimator in Eq. 9 and the estimator with the knowledge of true labels on the unlabeled pool.

Since our goal is improving the model uncertainty quantification quality on the unseen test data and also on the unlabeled pool so that it can improve the AF effectiveness of querying data points in AL, we next formally show the expected calibration error bound on the unlabeled pool and on the unseen test data of our AF by the theorem as follows:

**Theorem 4.2.** *For every round $t \in (0, T]$, given $m_t = m_{t-1} - k_t$ samples in the unlabeled pool $U_t$, $n_t = n_{t-1} + k_t$ samples in the queried dataset $S_t$, where $k_t$ is the number of selected samples from our AF, suppose the calibration error function is bounded by $L$, the calibration error estimator is unbiased, and the trained model is assumed to have $\frac{1}{k_t} \sum_{j \in n_{t-1}+[k_t]} \|\mathbb{E}[Y|h(x_j)] - h(x_j)\|_p^p \leq \epsilon$ over queried training points. Then, the expected calibration error on the unlabeled pool of Alg. 1 is also bounded by*

$$\frac{1}{m_t} \sum_{i=n_t+1}^{n_t+m_t} \|\mathbb{E}[Y|h(x_i)] - h(x_i)\|_p^p \leq \epsilon.$$

*Furthermore, for any $\delta \in (0, 1)$, with probability at least $1 - \delta$, the expected calibration error on the unseen test data of Alg. 1 is bounded by*

$$\mathbb{E}_{x \sim \mathbb{P}(X)} \|\mathbb{E}[Y|h(x)] - h(x)\|_p^p \leq \epsilon + \sqrt{\frac{L^2 \log(2/\delta)}{2(m_t + n_t)}}.$$

The proof is in Apd. A.2. Thm 4.2 suggests that when we query samples with the highest calibration error before leveraging DNN uncertainty, then train our model on these queried data points with a small enough calibration error loss, we can guarantee that the calibration error on the unlabeled pool and unseen data is also small. Furthermore, the more labeled samples $n_t$ and unlabeled samples $m_t$ we have, the tighter calibration error upper bound we can guarantee. We confirm this result in Fig. 5, where the ECE of our proposed method on the unlabeled pool is smaller than other baselines.

## 5 EXPERIMENTS

### 5.1 EXPERIMENTAL SETTINGS

**Baselines**. We follow Lüth et al. (2023) to compare with 6 main AF baselines, including **random** sampling, **least-confidence** from softmax, **Margin** (Wang & Shang, 2014), **BALD** (Gal et al., 2017), **Coreset** (Sener & Savarese, 2018), and **BADGE** (Ash et al., 2020). We also extend to compare with other AL settings, including data-augmentation (**CAMPAL** (Yang et al., 2023)), two-stage-based (**Rand-Entropy**, **Cluster-Margin** (Citovsky et al., 2021)), and batch-based (**BatchBALD** (Kirsch et al., 2019)). Since our method only modifies the AF, we maintain all baselines that share the same training strategy with the cross-entropy loss. Notably, we focus on the inductive AL (MacKay, 1992; Hübotter et al., 2024), do not use any semi, self-supervised (Gao et al., 2020; Bengar et al., 2021), or post-hoc recalibration techniques in training, though we still add them to compare in our ablation studies (e.g., in-processing calibration: **CALICO** (Querol et al., 2024); post-processing calibration: Temperature Scaling (TS), Adaptive TS (Balanya et al., 2024), **DDU** (Mukhoti et al., 2023), etc.).

**Other settings**. We deploy models across 6 standard datasets and backbones, including MNIST with MNIST-Net, Fashion-MNIST with GarmentClassifier, SVHN, balanced CIFAR-10, and imbalanced CIFAR-10-LT with ResNet-18, and ImageNet with ResNet-50. We use the train set as a warmup and an unlabeled pool, and the test set as an unseen test dataset. For all baselines, we use the same AL hyperparameters and other training hyperparameters. We run the results across 10 different random seeds, and these seeds are shared across baselines for a fair comparison. We evaluate models with accuracy and ECE (Naeini et al., 2015) (and also Kolmogorov-Smirnov (KS) error (Gupta et al., 2021) in Fig. 18). (Details are in Apd. B.1).

Table 1: Calibration error (lower is better) and accuracy (higher is better) comparison with different baselines across query times $t \in \{T/4, T/2, 3T/4, T\}$, where $T = 40$ on MNIST and $T = 100$ on other datasets. Scores are reported by mean $\pm$ standard deviation from 10 runs. Gray rows indicate our proposed method (i.e., Our **CUSAL** in Alg. 1). Best scores with the significant test are marked in **bold**. Figure details are in Fig. 8.

| Dataset | Method | Expected Calibration Error (↓) | | | | Accuracy (↑) | | | |
|---|---|---|---|---|---|---|---|---|---|
| | | $t = T/4$ | $t = T/2$ | $t = 3T/4$ | $t = T$ | $t = T/4$ | $t = T/2$ | $t = 3T/4$ | $t = T$ |
| MNIST | Random | 0.121 ± 0.025 | 0.079 ± 0.005 | 0.064 ± 0.002 | 0.058 ± 0.003 | 82.7 ± 3.1 | 89.8 ± 0.8 | 92.2 ± 0.2 | 93.2 ± 0.1 |
| | Least-conf | 0.123 ± 0.014 | 0.069 ± 0.011 | 0.039 ± 0.004 | 0.033 ± 0.006 | 83.1 ± 1.4 | 90.7 ± 1.2 | 94.4 ± 0.3 | 95.5 ± 0.6 |
| | Margin | 0.116 ± 0.018 | 0.068 ± 0.010 | 0.042 ± 0.003 | 0.038 ± 0.004 | 84.1 ± 1.5 | 90.9 ± 1.0 | 94.2 ± 0.2 | 95.1 ± 0.5 |
| | BALD | 0.109 ± 0.012 | 0.058 ± 0.007 | 0.036 ± 0.002 | 0.031 ± 0.003 | 83.3 ± 1.4 | 90.9 ± 1.1 | 94.5 ± 0.3 | 95.5 ± 0.6 |
| | Coreset | 0.122 ± 0.017 | 0.072 ± 0.008 | 0.047 ± 0.003 | 0.041 ± 0.004 | 82.9 ± 1.9 | 90.4 ± 1.0 | 93.7 ± 0.2 | 94.7 ± 0.4 |
| | BADGE | 0.111 ± 0.012 | 0.061 ± 0.008 | 0.038 ± 0.003 | 0.032 ± 0.005 | 85.3 ± 1.3 | 92.3 ± 0.7 | 94.8 ± 0.2 | 95.7 ± 0.4 |
| | Ours | **0.089** ± 0.014 | **0.046** ± 0.007 | **0.035** ± 0.003 | **0.030** ± 0.002 | **86.7** ± 1.6 | **93.3** ± 0.8 | **95.1** ± 0.2 | **95.9** ± 0.3 |
| SVHN | Random | 0.139 ± 0.004 | 0.119 ± 0.003 | 0.110 ± 0.004 | 0.103 ± 0.004 | 81.7 ± 0.3 | 84.8 ± 0.4 | 86.2 ± 0.4 | 87.6 ± 0.4 |
| | Least-conf | 0.134 ± 0.009 | 0.113 ± 0.006 | 0.099 ± 0.003 | 0.087 ± 0.003 | 82.6 ± 1.0 | 86.1 ± 0.7 | 88.0 ± 0.4 | 89.5 ± 0.3 |
| | Margin | 0.131 ± 0.004 | 0.110 ± 0.003 | 0.098 ± 0.002 | 0.089 ± 0.001 | 82.9 ± 0.4 | 86.3 ± 0.4 | 88.0 ± 0.2 | 89.4 ± 0.2 |
| | BALD | 0.132 ± 0.003 | 0.111 ± 0.003 | 0.100 ± 0.002 | 0.092 ± 0.002 | 82.6 ± 0.3 | 86.0 ± 0.4 | 87.6 ± 0.2 | 88.9 ± 0.2 |
| | Coreset | 0.133 ± 0.005 | 0.112 ± 0.003 | 0.100 ± 0.002 | 0.090 ± 0.002 | 82.6 ± 0.5 | 86.1 ± 0.4 | 87.8 ± 0.2 | 89.1 ± 0.2 |
| | BADGE | 0.129 ± 0.003 | 0.108 ± 0.003 | 0.096 ± 0.001 | 0.088 ± 0.001 | 83.2 ± 0.3 | 86.6 ± 0.4 | 88.3 ± 0.1 | 89.6 ± 0.1 |
| | Ours | **0.123** ± 0.002 | **0.103** ± 0.003 | **0.090** ± 0.001 | **0.081** ± 0.001 | **83.6** ± 0.4 | **86.9** ± 0.4 | **88.7** ± 0.2 | **90.1** ± 0.1 |
| F-MNIST | Random | 0.238 ± 0.007 | 0.210 ± 0.006 | 0.200 ± 0.006 | 0.200 ± 0.005 | 71.4 ± 0.6 | 75.6 ± 0.7 | 77.3 ± 0.6 | 77.9 ± 0.5 |
| | Least-conf | 0.248 ± 0.017 | 0.217 ± 0.017 | 0.203 ± 0.020 | 0.188 ± 0.016 | 71.6 ± 1.5 | 76.0 ± 1.6 | 77.9 ± 2.0 | 79.6 ± 1.6 |
| | BALD | 0.252 ± 0.019 | 0.203 ± 0.010 | 0.189 ± 0.012 | 0.175 ± 0.010 | 71.4 ± 1.8 | 77.7 ± 1.0 | 79.6 ± 1.2 | 80.8 ± 1.1 |
| | Margin | 0.238 ± 0.006 | 0.208 ± 0.006 | 0.196 ± 0.009 | 0.190 ± 0.005 | 72.2 ± 0.6 | 76.7 ± 0.6 | 78.5 ± 0.8 | 79.5 ± 0.5 |
| | Coreset | 0.244 ± 0.010 | 0.215 ± 0.010 | 0.202 ± 0.013 | 0.194 ± 0.009 | 71.6 ± 0.9 | 75.8 ± 1.4 | 77.7 ± 1.3 | 78.8 ± 0.9 |
| | BADGE | 0.233 ± 0.006 | 0.203 ± 0.004 | 0.191 ± 0.005 | 0.186 ± 0.004 | 72.8 ± 0.7 | 77.5 ± 0.4 | 79.3 ± 0.5 | 80.3 ± 0.4 |
| | Ours | **0.225** ± 0.008 | **0.190** ± 0.004 | **0.175** ± 0.004 | **0.165** ± 0.003 | **73.6** ± 1.1 | **78.7** ± 0.5 | **80.6** ± 0.4 | **81.9** ± 0.3 |
| CIFAR-10 | Random | 0.338 ± 0.003 | 0.312 ± 0.001 | 0.288 ± 0.003 | 0.287 ± 0.014 | 54.5 ± 0.6 | 59.7 ± 0.3 | 63.4 ± 0.2 | 63.8 ± 2.0 |
| | Least-conf | 0.345 ± 0.009 | 0.313 ± 0.007 | 0.285 ± 0.006 | 0.278 ± 0.007 | 54.9 ± 0.4 | 60.8 ± 0.5 | 64.7 ± 0.9 | 66.1 ± 0.6 |
| | Margin | 0.338 ± 0.002 | 0.309 ± 0.002 | 0.285 ± 0.002 | 0.276 ± 0.008 | 54.9 ± 0.2 | 60.8 ± 0.3 | 64.8 ± 0.7 | 66.0 ± 0.8 |
| | BALD | 0.338 ± 0.010 | 0.311 ± 0.013 | 0.283 ± 0.004 | 0.272 ± 0.007 | 54.8 ± 1.0 | 60.5 ± 0.5 | 64.7 ± 0.5 | 65.9 ± 0.5 |
| | Coreset | 0.343 ± 0.002 | 0.313 ± 0.003 | 0.287 ± 0.003 | 0.282 ± 0.008 | 54.7 ± 0.1 | 60.4 ± 0.3 | 64.1 ± 0.5 | 65.0 ± 1.1 |
| | BADGE | 0.334 ± 0.006 | 0.304 ± 0.003 | 0.282 ± 0.003 | 0.270 ± 0.008 | **55.2** ± 0.5 | **61.1** ± 0.2 | **65.4** ± 0.9 | **67.0** ± 0.7 |
| | Ours | **0.329** ± 0.009 | **0.300** ± 0.004 | **0.279** ± 0.004 | **0.261** ± 0.006 | **55.6** ± 0.9 | **61.5** ± 0.5 | **65.1** ± 0.4 | **67.0** ± 0.4 |

Figure 4: 2-D visualizations of AL performance regarding ECE (x-axis) and Accuracy (y-axis) on CIFAR-10 from Tab. 1. **Our method is closest to the Best performance point (i.e., 100% accuracy and 0.0 ECE).**

## 5.2 MAIN RESULTS

Tab. 1 shows results at particular time steps $t \in \{T/4, T/2, 3T/4, T\}$ and Fig. 8 summarizes results across $T$ time horizons. It can be seen from these results that our **CUSAL** method consistently outperforms other baselines at almost every time step $t \in [T]$ across different settings by lower ECE. Accompanied by the accuracy performance, even in a challenging dataset like CIFAR-10, our method is the closest point to the best performance point in Fig. 4. This implies that our AF can query more informative samples and needs fewer query costs to achieve the desired behavior in AL.

In particular, on **Fashion (F)-MNIST**, we observe a remarkably lower ECE and higher accuracy from our AF when compared to other baselines. For example, ours is at around 0.190, 0.175, and 0165 in ECE, lower than others by more than 0.013, 0.014 and 0.010 at time step $t = \{50, 75, 100\}$ respectively. Similarly, it has a 78.7%, 80.6%, and 81.9% accuracy, higher than others by more than 1% at each $t = \{50, 75, 100\}$ correspondingly. In addition to ECE, we also evaluate calibration error based on Kolmogorov-Smirnov (KS) error, a binning-free calibration metric in Fig. 18. We observe that the calibration error across ECE and KS metrics is consistent, and our method outperforms other baselines by lower ECE, KS error, and higher accuracy. It is also worth noting that the least-confident method is uncalibrated, leading to uninformative query samples and resulting in a similarly poor result to random sampling. Meanwhile, methods that aim to improve uncertainty quality, like BALD and BADGE, can improve the calibration and generalization. This confirms the correlation

between high-quality uncertainty estimation and generalization in AL. Another grayscale image dataset is **MNIST**, at $t = 10$, our method also achieves around 0.089 ECE, significantly better than other baselines by more than 0.02. This demonstrates that our Alg. 1 can help improve the uncertainty estimation quality of the DNN in the AL task. In addition to uncertainty estimation, our method also achieves the highest accuracy with 86.7%, illustrating that a calibrated uncertainty sampling method can help to query more informative samples to enhance generalization.

Since MNIST is class-balanced, we next evaluate on **SVHN** to test performance on an imbalanced class distribution (more results with CIFAR-10-LT and data-augmentation are in Apd. B.3.2). We also observe similar behaviors on this real-world dataset, where our method is notably better calibrated and more accurate than other baselines. For instance, our proposed method achieves 0.090 and 0.081 ECE, lower than others by more than 0.006 at $t = \{75, 100\}$. At the same time, it also reaches 88.7% and 90.1%, higher than others by more than 0.5%.

Regarding a more complex setting like **CIFAR-10**, although Fig. 4 shows that all of the methods have difficulty improving in this setting, our proposed AF has a competitive result with BADGE in accuracy and still brings out better performance with 61.5%, 65.1%, and 67%, higher than others by more than 0.7%, 0.4%, and 0.9% at time $t = \{50, 75, 100\}$ respectively. Notably, at the same time, it still outperforms other baselines in uncertainty estimation quality with 0.300, 0.279, and 0.261 ECE, lower than all others by more than 0.004, 0.003, and 0.009 correspondingly. This suggests that even when models cannot perfectly classify samples, our proposed AF still can help improve the calibration quality, signaling when it is likely to be true or wrong in the real world.

Finally, we provide results on a larger-scale setting with **ImageNet**. From Fig. 6 and Tab. 2, we observe that our method achieves lower ECE with 0.2868, 0.2523, 0.2301, and 0.2228 and higher accuracy with 56.16%, 59.04%, 60.28%, and 61.00% at $n_t = \{40k, 60k, 80k, 100k\}$ (i.e., $t = \{2, 4, 6, 8\}$) than other query-based baselines. Notably, we also compare with CAMPAL (Yang et al., 2023). It is worth noting that CAMPAL modifies the training strategy by using data augmentation techniques, while our setting does not modify the training strategy. Therefore, we add a new setting where our proposed AF is combined with the data-augmentation strategy in CAMPAL. We observe that this combination also has a better performance in both accuracy and uncertainty estimation quality than CAMPAL$_{\text{BADGE}}^{\text{CHAMBER}}$, confirming the effectiveness of our AF for trustworthiness in AL.

## 5.3 ABLATION STUDIES

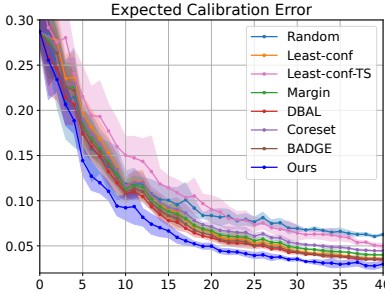 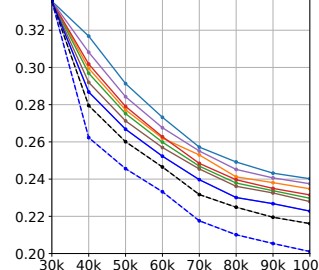 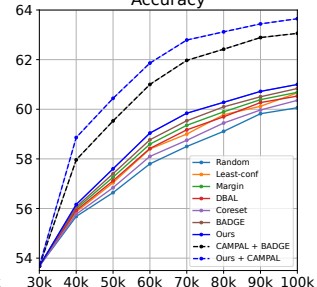

Figure 5: Calibration quality on the unlabeled pool with MNIST. More results are in Tab. 4 and Fig. 11.

Figure 6: Calibration and accuracy comparison across the number of queried samples, equivalently at query times $t \in [T]$, where $T = 8$ on a large-scale setting with ImagetNet. Table details are in Tab. 2.

**The quality of our calibration error estimator**. Recall our Eq. 9 estimates the calibration error per sample on the unlabeled pool. To evaluate this estimator's quality, we compare it to the estimator where we know the true label on the unlabeled pool of Popordanoska et al. (2022) in Eq. 7. From Fig. 3, we observe that when the number of cumulative queried samples increases, the gap between the two estimators decreases correspondingly. This shows that the quality of our estimator in Eq. 9 improves, and confirms the dependence on the number of queried samples in Thm. 4.1.

**Our uncertainty estimation quality on the unlabeled pool**. A calibrated DNN on the unlabeled pool is a necessary condition for a high-quality uncertainty sampling AF in AL. Therefore, to better understand why our proposed AF can bring out a better generalization performance, we compare the calibration on the unlabeled pool. Fig. 5 shows that our method consistently outperforms other baselines across every $t \in [T]$. This means that our AF helps the model improve its predictive

certainty, and we can leverage DNN uncertainty to better query data points to enhance AL performance. Furthermore, this result also confirms that, by selecting the least calibrated samples to query, our designed strategy can help improve calibration on the unlabeled pool as seen in Thm. 4.2.

**Comparison with querying by least-calibrated only**. Recall our Alg. 1 query by the lexicographic order, prioritizing the least calibrated, then the least certain samples. To test the effectiveness of this lexicographic ordering, we also compare our AF with a method of only querying the least calibrated samples. Fig. 13 shows that our AF results in a better performance with a lower ECE and higher accuracy, confirming the effectiveness of the lexicographic order during query time.

**Frequency of samples selected based on calibration versus uncertainty.** Our lexicographic order enables our method to focus on querying samples with calibration errors in early rounds, then shift to uncertainty sampling as calibration improves in later rounds. Specifically, from Tab. 5, the number of selected samples based on calibration error is high at early rounds. The model can then improve the calibration performance. As a result, in later rounds, the calibration error is lower and more uniform across samples (Fig. 3, Fig. 11), resulting in more samples being selected by uncertainty-sampling.

### 5.4 COMPARISON WITH RECALIBRATION METHODS

To improve predictive performance with calibration in AL, a simple solution may be to use a post-hoc recalibration step during training after each query time. Hence, we additionally compare with this setting in Fig. 5, Fig. 7, and Fig. 12. Specifically, Least-confident-TS, C-Margin-TS, and C-Margin-ATS are the result of Least-confident and Cluster-Margin, where we split 80% of the cumulative data to train and 20% to do post-hoc temperature scaling (TS) and adaptive-TS (ATS) to recalibrate the trained model (Balanya et al., 2024). C-Margin-TS (on trainset) is using TS directly on 100% of the cumulative data. We also compare with other modern calibration AL baselines (e.g., post-processing calibration DDU (Mukhoti et al., 2023) and in-processing calibration CALICO (i.e., optimize with a recalibration regularizer in training) (Querol et al., 2024).

Firstly, we observe that our method consistently outperforms other post-hoc recalibration methods in this AL setting. Specifically, compared with methods that apply recalibration on hold-out datasets (e.g., Least-confident-TS in Fig. 5, C-Margin-TS, and C-Margin-ATS in Fig. 7), we have a significantly lower ECE and higher accuracy. This is because such methods require splitting the dataset, thereby sacrificing samples in training, which results in a significantly lower accuracy. Meanwhile, the calibration performance depends on model confidence and accuracy. As a result, the ECE of this method on both the unlabeled pool and unseen test datasets is also significantly worse than our method across AL time horizons. Secondly, if we apply TS directly on 100% of the cumulative data, from Fig. 7, we can see there is no difference between "C-Margin" and "C-Margin-TS (on trainset)", proving that using TS on the training data does not help improve calibration. This is because using calibration algorithms on training data introduces bias (since the labeled data is already biased in AL and has a distribution shift from the original data distribution). Finally, regarding comparison with modern calibration AL baselines in Fig. 12 and Fig. 7, the poor performance of DDU is similar to that of other post-hoc recalibration baselines because it also requires a hold-out recalibration set. Regarding CALICO, this in-processing method leads to performance degradation due to the training bias and a tradeoff between prediction power and calibration regularizer (Querol et al., 2024).

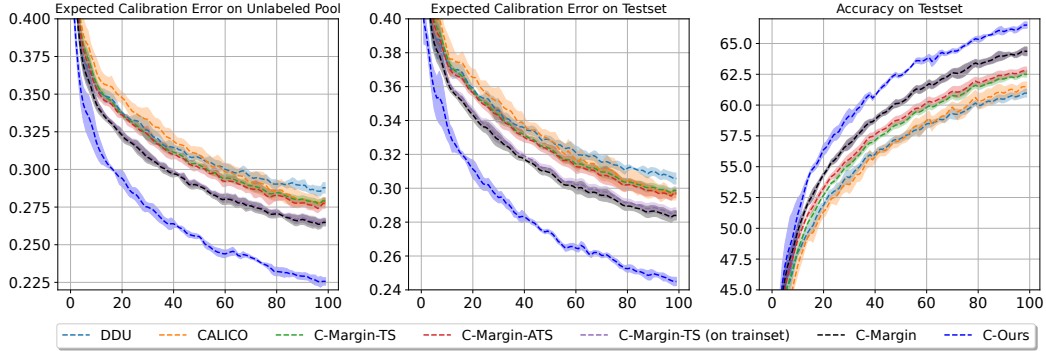

Figure 7: Expected Calibration error (lower is better) and accuracy (higher is better) comparison with different recalibration methods across query times $t \in [T]$, where $T = 100$ on CIFAR-10-LT.

## 6 RELATED WORK

The literature on the pool-based AL can be categorized into innovating AF and innovating training loss functions (Werner et al., 2024). In particular, innovating AF focuses on designing a new AF and fixing the training strategy. On the other hand, innovating on the training loss function additionally tries to change the training strategy, such as using data augmentation (Kim et al., 2021; Bengar et al., 2021) or leveraging samples from the unlabeled pool (Lüth et al., 2023; Gao et al., 2020).

In the scope of this paper, we consider the innovating AF setting. The literature on this domain could be sub-categorized into two main approaches: *uncertainty-based sampling* and *diversity-based sampling*. Regarding *uncertainty-based sampling*, its main idea is querying the most uncertain samples from the model, including querying samples that lie closest to the linear decision boundary (Schohn & Cohn, 2000; Balcan et al., 2006) or using the DNN predictive model uncertainty, e.g., softmax layer (least-confident), using its predictive entropy, and using the smallest separation of the top two class predictions (margin sampling) (Wang & Shang, 2014). Later on, Gal et al. (2017) proposes BALD by using MC-Dropout to improve the DNN uncertainty estimation quality on the unlabeled pool, resulting in generalization improvement. Regarding *diversity-based sampling*, its main idea is to query the most diverse samples. Common approaches include clustering and selecting unlabeled samples that are the furthest away from all cluster centers (Coreset) (Sener & Savarese, 2018; Geifman & El-Yaniv, 2017), selecting unlabeled samples that are maximally indistinguishable (Gissin & Shalev-Shwartz, 2019). Following this direction, Ash et al. (2020) introduces BADGE by sampling groups of points that are disparate and the high magnitudes when represented in a hallucinated gradient space. This can be seen as a hybrid version of the uncertainty-based and diversity-based.

Although the aforementioned AL techniques have shown promising generalization results, their performance regarding the quality of uncertainty estimation remains in question. Meanwhile, this uncertainty quality is an important aspect of a reliable ML model, especially in the AL setting. Hence, there has been a growing interest in studying the correlation between calibration and AL recently (Sürer & Wild, 2024; Thomas-Mitchell et al., 2023; Rožanec et al., 2023). Closest to our work is CALICO (Querol et al., 2024), a calibrated framework for AL that jointly trains cross-entropy loss with an energy-based function. Yet, its improvement over Least-conf is minor, as the AF stays the same and the joint training is in a semi-supervised way (see Fig. 13). DDU (Mukhoti et al., 2023) also evaluates calibration, but requires an additional hold-out dataset to do post-hoc recalibration techniques. Hence, its performances are close to Least-conf-TS (see Fig. 12). In contrast, our method enhances uncertainty estimation and generalization performance without requiring modifications to the loss function or using any hold-out recalibration in training.

Regarding consistent calibration error estimators, Zhang et al. (2020) and Popordanoska et al. (2022) propose estimators but require the true label under the IID assumption. Lately, (Popordanoska et al., 2024) tackles this limitation by introducing a consistent estimator without labels under label-shift. There also exists a covariate-shift version in the domain adaptation (Popordanoska et al., 2023), but without theoretical verification for the proposed estimator. In contrast, our work proposes a provable consistent estimator on the unlabeled pool under the setting of covariate shift in the AL framework.

## 7 CONCLUSION

AF based on uncertainty sampling has become a standard approach in pool-based AL. Yet, the uncertainty quantification quality derived from the DNN model is often uncalibrated. This leads to the model not only being unreliable on unseen data but also to selecting non-informative samples from the unlabeled pool. Hence, we propose **CUSAL**, a reliable AF, aiming to enhance both calibration and generalization. Our novel AF uses the lexicographic order for calibration and uncertainty sampling, prioritizing fixing calibration error first by using a kernel calibration estimator to choose the least calibrated samples in the unlabeled pool. Theoretically, we provide the calibration estimator error, expected calibration error bound on the unlabeled pool, and on the unseen test set of our AF. Our proposed method empirically surpasses other baselines by having a lower calibration and generalization error across pool-based AL. With this result, we hope our work can open a new direction for improving trustworthiness in the AL setting. Future work includes tackling the computational efficiency limitation of our kernel estimator and extending it to Large language Models.

**Reproducibility**. The source code to reproduce our results is available in the supplementary material zipped file. In Apd. A, we provide the proofs for all our theoretical results, including: proof of Theorem 4.1 in Apd. A.1; proof of Theorem 4.2 in Apd. A.2. In Apd. B, we provide additional information about our experiments, including: sufficient details about experimental settings in Apd. B.1; demo code in Apd. B.2; additional results in Apd. B.3 with detailed comparison to other baselines in Apd. B.3.1; evaluation on an imbalanced benchmark in B.3.2; calibration error on the unlabeled pool in Apd. B.3.3; additional ablation studies in Apd. B.3.4.

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

# CALIBRATED UNCERTAINTY SAMPLING FOR ACTIVE LEARNING (SUPPLEMENTARY MATERIAL)

In this appendix, we collect proofs and remaining materials deferred from the main paper. In Appendix A, we provide the proofs for all our theoretical results, including: proof of Theorem 4.1 in Appendix A.1; proof of Theorem 4.2 in Appendix A.2. In Appendix B, we provide additional information about our experiments, including: details about experimental settings in Appendix B.1; demo code in Appendix B.2; additional results in Appendix B.3 with detailed comparison to other baselines in Appendix B.3.1; evaluation on an imbalanced benchmark in B.3.2; calibration error on the unlabeled pool in Appendix B.3.3; additional ablation studies in Appendix B.3.4. Finally, the source code to reproduce our results is available in the zipped file of this supplementary material.

## A PROOFS

### A.1 PROOF OF THEOREM 4.1

*Proof.* Recall the covariate shift of AL existing between the cumulative labeled data $S$ and samples from the unlabeled pool $U$ in Section 4, i.e.,

$$S \sim \tilde{\mathbb{P}}(X)\mathbb{P}(Y|X), \quad U \sim \tilde{\mathbb{P}}(X_S)\mathbb{P}(Y|X). \tag{15}$$

For the conditional expectation, we have

$$\mathbb{E}_{\pi_U}[Y|h(X) = h(x)] = \sum_{y_k \in \mathcal{Y}} y_k \frac{p_u(Y = y_k, h(X) = h(x))}{p_u(h(X) = h(x))} \tag{16}$$

$$= \sum_{y_k \in \mathcal{Y}} y_k \frac{p_u(Y = y_k|h(X) = h(x))p_u(h(X) = h(x))}{p_u(h(X) = h(x))}. \tag{17}$$

By the covariate shift in Equation 15, thus we get

$$\mathbb{E}_{\pi_U}[Y|h(X) = h(x)] = \sum_{y_k \in \mathcal{Y}} y_k \frac{p_s(Y = y_k|h(X) = h(x))p_s(h(X) = h(x))}{p_s(h(X) = h(x))} \tag{18}$$

$$= \sum_{y_k \in \mathcal{Y}} y_k \frac{p_s(h(X) = h(x)|Y = y_k)p_s(Y = y_k)}{p_s(h(X) = h(x))}. \tag{19}$$

As mentioned in the covariate shift in AL (Liu et al., 2015; Fannjiang et al., 2022), for every round $t \in [T]$, the queried samples $\{(x_i, y_i)\}_{i=n_{t-1}+1}^{n_t}$ are sampled IID w.r.t. $\mathbb{P}_s(X, Y) = \tilde{\mathbb{P}}(X)\mathbb{P}(Y|X)$, when $h(x)$ is a probabilistic acquisition function. Therefore, suppose $n_t$ samples in the cumulative labeled set are sampled IID w.r.t. $\mathbb{P}_s(X, Y)$, then with $p_s$ is the corresponding probability density for $\mathbb{P}_s(X, Y)$, applying the result of Popordanoska et al. (2022), we obtain that when $p_{h(x)}h(x)$ is Lipschitz continuous over the interior of the simplex, $\exists$ a kernel $k$ s.t.

$$\plim_{n_t \to \infty} \frac{\sum_{i=1}^{n_t} k\left(h(x); h(x_i)\right) y_i}{\sum_{i=1}^{n_t} k\left(h(x); h(x_i)\right)} = \sum_{y_k \in \mathcal{Y}} y_k \frac{p_s(h(X) = h(x)|Y = y_k)p_s(Y = y_k)}{p_s(h(X) = h(x))} \tag{20}$$

$$= \mathbb{E}_{\pi_U}[Y|h(x)], \tag{21}$$

i.e., our estimator in Equation 9 is a point-wise consistent estimator under AL with covariate shift.

On the other hand, let us denote $s := h(x)$, $\hat{f}_{n_t,b}(s) := \mathbb{E}_{\pi_U}\left[\widehat{Y|h(x)}\right]$, and $f(s) := \mathbb{E}_{\pi_U}[Y|h(x)]$, then, we have

$$\mathbb{E}\left[\left|\mathbb{E}_{\pi_U}\left[\widehat{Y|h(x)}\right] - \mathbb{E}_{\pi_U}[Y|h(x)]\right|^2\right] = \mathbb{E}\left[\left|\hat{f}_{n_t,b}(s) - f(s)\right|^2\right] \tag{22}$$

$$= Var\left(\hat{f}_{n_t,b}(s)\right) + \left[Bias\left(\hat{f}_{n_t,b}(s)\right)\right]^2. \tag{23}$$

For $K$ classes, bandwidth $b$, with $k(\cdot)$ is a Dirichlet kernel in the form of Equation 10, following the result of Ouimet & Tolosana-Delgado (2022), we have the bias is as follows

$$Bias\left(\hat{f}_{n_t,b}(s)\right) = bg(s) + \mathcal{O}(b), \tag{24}$$

where

$$g(s) := \sum_{i\in[K]} (1 - (K+1)s_i)\frac{\partial}{\partial s_i}f(s) + \frac{1}{2}\sum_{i,j\in[K]} s_i\left(\mathbb{I}_{i=j} - s_j\right)\frac{\partial^2}{\partial s_i\partial s_j}f(s). \tag{25}$$

And the variance is as follows

$$Var\left(\hat{f}_{n_t,b}(s)\right) = n_t^{-1}A_b(s)\left(f(s) + \mathcal{O}(b^{1/2})\right) - \mathcal{O}(n_t^{-1}), \tag{26}$$

where

$$A_b(s) := \begin{cases} b^{-K/2}\psi(s)(1 + \mathcal{O}_s(b)), & \text{if } s_i/b \to \infty, \forall i \in [K], \\ & \text{and } (1 - ||s||_1)/b \to \infty, \\ \\ b^{-(K+|\mathcal{J}|)/2}\psi_{\mathcal{J}}(s)\prod_{i\in\mathcal{J}}\frac{\Gamma(2k_i+1)}{2^{2k_i+1}\Gamma^2(k_i+1)}\left(1 + \mathcal{O}_{k,s}(b)\right), & \text{if } s_i/b \to k_i, \forall i \in \mathcal{J}, \\ & s_i/b \to \infty, \forall i \in [K]\setminus\mathcal{J}, \\ & \text{and } (1 - ||s||_1)/b \to \infty, \end{cases} \tag{27}$$

where

$$\psi(s) := \psi(s) \text{ and } \psi_{\mathcal{J}}(s) := \left[(4\pi)^{K-|\mathcal{J}|}(1 - ||s||_1)\prod_{i\in[K]\setminus\mathcal{J}} s_i\right]^{-1/2}, \tag{28}$$

for every subset of indices $\mathcal{J} \in [K]$. Plug this bias and variance result into Equation 22, we obtain the mean square error of our estimator in Equation 7 is bounded by

$$\mathbb{E}\left[\left|\hat{f}_{n_t,b}(s) - f(s)\right|^2\right] = n_t^{-1}A_b(s)\left(f(s) + \mathcal{O}(b^{1/2})\right) - \mathcal{O}(n_t^{-1}) + b^2g^2(s) + \mathcal{O}(b^2) \tag{29}$$

$$= n_t^{-1}b^{-K/2}\left(\psi(s)f(s) + \mathcal{O}_s(b^{3/2})\right) - \mathcal{O}(n_t^{-1}) + b^2g^2(s) + \mathcal{O}(b^2) \tag{30}$$

$$= n_t^{-1}b^{-K/2}\psi(s)f(s) + b^2g^2(s) + \mathcal{O}_s(n_t^{-1}b^{\frac{-K+1}{2}}) + \mathcal{O}(b^2) - \mathcal{O}(n_t^{-1}) \tag{31}$$

$$\le \mathcal{O}(n_t^{-1}b^{\frac{-K+1}{2}}) + \mathcal{O}(b^2) \tag{32}$$

of Theorem 4.1. $\qquad\square$

## A.2 PROOF OF THEOREM 4.2

*Proof.* Given our setting in Section 2, for every round $t \in [T]$, we have the union of $n_t$ samples from the queried set and $m_t$ samples from the unlabeled pool is fixed. As discussed in Section 4, this total $m_t + n_t$ samples, i.e., $\{x_i, y_i\}_{i\in[m_t+n_t]}$ are also drawn IID from $\mathbb{P}(X, Y)$. So, recall Hoeffding's inequality, i.e., let $Z_1, \cdots, Z_n$ be independent bounded random variables with $Z_i \in [a, b]$ for all $i$, where $-\infty < a < b < \infty$. Then

$$\mathbb{P}\left(\frac{1}{n}\sum_{i=1}^{n}(Z_i - \mathbb{E}[Z_i]) \ge t\right) \le \exp\left(\frac{-2nt^2}{(b-a)^2}\right). \tag{33}$$

Firstly, following conditions in the theorem, by the selection of our AF, i.e., selecting $k_t$ highest calibration error samples from $m_t + k_t$ samples in the unlabeled pool, we have

$$\frac{1}{m_t}\sum_{i=n_t+1}^{n_t+m_t} ||\mathbb{E}[Y|h(x_i)] - h(x_i)||_p^p \le \frac{1}{m_t + k_t}\sum_{i=n_{t-1}+1}^{n_{t-1}+k_t+m_t} ||\mathbb{E}[Y|h(x_i)] - h(x_i)||_p^p \tag{34}$$

$$\le \frac{1}{k_t}\sum_{j=n_{t-1}+1}^{n_{t-1}+k_t} ||\mathbb{E}[Y|h(x_j)] - h(x_j)||_p^p. \tag{35}$$

Combining with the assumption $\frac{1}{k_t} \sum_{j=n_{t-1}+1}^{n_{t-1}+k_t} \|\mathbb{E}[Y|h(x_j)] - h(x_j)\|_p^p \leq \epsilon$, we obtain the expected calibration error on the unlabeled pool of Algorithm 1 is also bounded by

$$\frac{1}{m_t} \sum_{i=n_t+1}^{n_t+m_t} \|\mathbb{E}[Y|h(x_i)] - h(x_i)\|_p^p \leq \frac{1}{m_t+k_t} \sum_{i=n_{t-1}+1}^{n_{t-1}+k_t+m_t} \|\mathbb{E}[Y|h(x_i)] - h(x_i)\|_p^p \leq \epsilon. \quad (36)$$

On the other hand, since the empirical calibration error and the expected error on the unseen test data are

$$\frac{1}{m_t+n_t} \sum_{i=1}^{m_t+n_t} \|\mathbb{E}[Y|h(x_i)] - h(x_i)\|_p^p \quad \text{and} \quad \mathbb{E}_{x\sim\mathbb{P}(X)}\|\mathbb{E}[Y|h(x)] - h(x)\|_p^p, \text{ respectively.} \quad (37)$$

So, applying Hoeffding's inequality, we have

$$\mathbb{P}\left(\left|\mathbb{E}_{x\sim\mathbb{P}(X)}\|\mathbb{E}[Y|h(x)] - h(x)\|_p^p - \frac{1}{m_t+n_t} \sum_{i=1}^{m_t+n_t} \|\mathbb{E}[Y|h(x_i)] - h(x_i)\|_p^p\right| \geq t\right)$$

$$\leq 2\exp\left(\frac{-2(n_t+m_t)t^2}{L^2}\right). \quad (38)$$

Let $\delta = 2\exp\left(\frac{-2(m_t+n_t)\cdot t^2}{L^2}\right)$, equivalently $t = \sqrt{\frac{L^2\log(2/\delta)}{2(m_t+n_t)}}$, i.e.,

$$\mathbb{P}\left(\left|\mathbb{E}_{x\sim\mathbb{P}(X)}\|\mathbb{E}[Y|h(x)] - h(x)\|_p^p - \frac{1}{m_t+n_t} \sum_{i=1}^{m_t+n_t} \|\mathbb{E}[Y|h(x_i)] - h(x_i)\|_p^p\right| \leq \sqrt{\frac{L^2\log(2/\delta)}{2(m_t+n_t)}}\right)$$

$$\geq 1 - \delta. \quad (39)$$

Due to

$$\frac{1}{m_t+n_t} \sum_{i=1}^{m_t+n_t} \|\mathbb{E}[Y|h(x_i)] - h(x_i)\|_p^p \quad (40)$$

$$= \frac{1}{m_t+n_t} \left[\sum_{i=n_t+1}^{n_t+m_t} \|\mathbb{E}[Y|h(x_i)] - h(x_i)\|_p^p + \sum_{j=1}^{n_t} \|\mathbb{E}[Y|h(x_j)] - h(x_j)\|_p^p\right] \quad (41)$$

$$= \frac{1}{m_t+n_t} \left[\sum_{i=n_t+1}^{n_t+m_t} \|\mathbb{E}[Y|h(x_i)] - h(x_i)\|_p^p + \sum_{i=1}^{t} \sum_{j=n_{t-1}+1}^{n_{t-1}+k_t} \|\mathbb{E}[Y|h(x_j)] - h(x_j)\|_p^p\right] \quad (42)$$

$$\leq \frac{1}{m_t+n_t} \left[\sum_{i=n_t+1}^{n_t+m_t} \|\mathbb{E}[Y|h(x_i)] - h(x_i)\|_p^p + \epsilon \sum_{i=1}^{t} k_t\right] \quad (43)$$

$$\leq \frac{m_t\epsilon + n_t\epsilon}{m_t+n_t} \left(\text{by Equation 36 and } \sum_{i=1}^{t} k_t \leq n_t\right), \quad (44)$$

we obtain

$$\mathbb{P}\left(\mathbb{E}_{x\sim\mathbb{P}(X)}\|\mathbb{E}[Y|h(x)] - h(x)\|_p^p \leq \epsilon + \sqrt{\frac{L^2\log(2/\delta)}{2(m_t+n_t)}}\right) \geq 1 - \delta, \quad (45)$$

i.e., with probability at least $1 - \delta$, the expected calibration error on the unseen data of Algorithm 1 is bounded by

$$\mathbb{E}_{x\sim\mathbb{P}(X)}\|\mathbb{E}[Y|h(x)] - h(x)\|_p^p \leq \epsilon + \sqrt{\frac{L^2\log(2/\delta)}{2(m_t+n_t)}} \quad (46)$$

of Theorem 4.2. $\qquad\square$

# B    Experimental Details

## B.1    Experimental settings

**AL hyper-parameter settings.** We set the number of warm-up datapoints $n_0 = 20$ on MNIST & F-MNIST (Gal et al., 2017), $n_0 = 500$ on SVHN, $n_0 = 1000$ on CIFAR-10 & CIFAR-10-LT (Werner et al., 2024), and $n_0 = 30000$ on ImageNet. Following Gal et al. (2017). All warm-up datasets are sampled randomly but balanced (i.e., the number of samples per class is equal), except the imbalanced CIFAR-10-LT experiments in Apd. B.3.2. For feasible computation times, we set the number of time horizons (i.e., AL rounds) $T = 8$ on ImageNet, $T = 40$ on MNIST, and $T = 100$ on other datasets. For each round $t \in [T]$, we set the number of queried samples $k = 10$ on MNIST & F-MNIST, $k = 50$ on SVHN, $k = 100$ on CIFAR-10 & CIFAR-10-LT, and $k = 10000$ on ImageNet.

**Dataset & other hyper-parameters**. We deploy the models on six datasets, including the MNIST (& Fashion-MNIST) dataset with $d = 28 \times 28$ digit handwriting (& Zalando's article) images in $K = 10$ classes; the SVHN with $d = 32 \times 32 \times 3$ house numbers images in $K = 10$ classes; the CIFAR-10 (& CIFAR-10-LT) dataset with $d = 32 \times 32 \times 3$ images in $K = 10$ classes; the ImageNet (ILSVRC 2012) dataset with $d = 244 \times 244 \times 3$ image dimensions in $K = 1000$ classes. Regarding the model architectures and hyper-parameters settings, we use MNIST-Net for MNIST (Lecun et al., 1998), GarmentClassifer for Fashion-MNIST (Xiao et al., 2017), RestNet-18 (He et al., 2016) for SVHN (Netzer et al., 2011), CIFAR-10 (Krizhevsky & Hinton, 2009), and CIFAR-10-LT (Cui et al., 2019), and RestNet-50 (He et al., 2016) for ImageNet (Deng et al., 2009). We use the Adam optimizer with the Cross-entropy loss, learning rate equals 0.001, batch sizes equal 128, the number of training epochs equals 30 on MNIST, SVHN, CIFAR-10, CIFAR-10-LT, and ImageNet, 60 on Fashion-MNIST. We only normalize data in the data processing step, except in the data-augmentation experiments with CAMPAL (Yang et al., 2023). To evaluate models, we use accuracy and ECE (bin size equals 10) metrics. Regarding our calibration error estimator hyper-parameters in Equation 9, we use the absolute norm $p = 1$ and set the bandwidth of the Dirichlet kernel $b = 0.001$.

**Baseline details**: Regarding our baseline comparison in the main paper, we compare with other AF (i.e., query-based) methods and follow (Lüth et al., 2023) to select the baselines. To the best of our knowledge, Lüth et al. (2023); Werner et al. (2024) have shown that the following baselines are still state-of-the-art across query-based methods in AL, including:

- *Random*: queries unlabeled samples randomly from the unlabeled pool, i.e., $x_t^* = \arg\max_{x \in U_t} A(x; \theta_{t-1}) = \mathbb{U}$, where $\mathbb{U}$ represents the uniform distribution.

- *Least-confident* (Roy & McCallum, 2001): queries unlabeled samples from the unlabeled pool by the least confident (i.e., most uncertain) samples of the DNN, i.e., $x_t^* = \arg\max_{x \in U_t} \left(1 - \max_{y \in [K]}[h(x)]_y\right)$.

- *Least-confident-TS* (Guo et al., 2017): queries similar to the Least-confident strategy. However, the DNN is calibrated with an additional hold-out validation dataset by the temperature scaling technique.

- *Rand-Entropy*: is a two-stage-based by querying $k_m$ samples by using *Random*, then selecting $k_t$ with the highest predictive entropy from $k_m$ samples.

- *Margin* (Wang & Shang, 2014): queries unlabeled samples that are closest to the decision boundary (the "margin") of the trained classifier according to a distance function between differences of two most confident classes.

- *Cluster-Margin* (Citovsky et al., 2021): is a two-stage-based and an extension of *Margin* by querying $k_m$ samples by using the margin, then selecting $k_t$ from $k_m$ samples by a cluster-based approach to improve diversity.

- *BALD* (Gal et al., 2017): queries by DNN uncertainty, but the model uncertainty is obtained by sampling the probabilistic model's output with Monte-Carlo dropout sampling through the lens of the Bayesian view (Gal & Ghahramani, 2016).

- *BatchBALD* (Kirsch et al., 2019): is an extension of *BALD* by jointly scoring points by estimating the mutual information between a joint of data points and the model parameters.

- *Coreset* (Sener & Savarese, 2018): queries by diversity, i.e., unlabeled samples that are the furthest away from all cluster centers. The clusters are obtained by applying K-Means on a semantically meaningful space with encoded data from the DNN classifier.

- *BADGE* (Ash et al., 2020): queries by incorporating both DNN uncertainty and sample diversity of every selected batch. In particular, it employs a variant of K-Means to select disparate groups of points. Additionally, it uses gradient embeddings of unlabeled samples to query samples with the highest magnitude when represented in a hallucinated gradient space, i.e., samples that the model is expected to change much to improve its performance.

**Source code and computing systems**. Our source code includes the dataset scripts, setup for the environment, and our provided code (details in README.md). We run our code on a single GPU: NVIDIA RTX A6000-49140MiB with 8-CPUs: AMD Ryzen Threadripper 3960X 24-Core with 8GB RAM per each and require 160GB available disk space for storage.

## B.2 DEMO NOTEBOOK CODE FOR ALGORITHM 1

```python
import torch

#Estimate the calibration error of each sample by Eq.9.
def get_ratio_canonical_per_samples(f, y, bandwidth, p = 1):
    log_kern = get_kernel(f, bandwidth, device)
    kern = torch.exp(log_kern)
    y_onehot = nn.functional.one_hot(y, num_classes=f.shape[1])
    kern_y_splits = kern[y.shape[0]:, :y.shape[0]]
    kern_y = torch.matmul(kern_y_splits, y_onehot)
    den = torch.sum(kern_y_splits, dim=1)
    den = torch.clamp(den, min=1e-10)
    ratio = kern_y / den.unsqueeze(-1)
    ce_per_samples = torch.sum(torch.abs(ratio - f[y.shape[0]:])**p, dim=1)
    return ce_per_samples

#Select top-k samples by the lexicographic order in Eq.12.
def sampling(model, pool_loader, select_samples, trainloader):
    model.eval()
    train_outputs, train_targets = [], []
    with torch.no_grad():
        for data, target in train loader:
            output = model(data)
            output = torch.softmax(output, dim=1)
            train_outputs = torch.cat((train_outputs, output), 0)
            train_targets = torch.cat((train_targets, target), 0)
    outputs = []
    with torch.no_grad():
        for data, target in pool_loader:
            output = model(data)
            output = torch.softmax(output, dim=1)
            outputs = torch.cat((outputs, output), 0)

    input_f = torch.cat((train_outputs, outputs_1), 0)
    ece = get_ratio_canonical_per_samples(input_f,train_targets,bandwidth=0.001)
    conf = outputs.max(1).values.numpy()
    out = np.column_stack((conf, ece))
    return np.lexsort((out[:,0],-out[:,1]))[:select_samples]

if __name__ == "__main__"
    net = MNIST_Net()
    pool_data, pool_labels = np.load(data_path)
    idxs_unlabeled = sampling_balance(pool_labels)
    for t in range(T):
        selected_data = pool_data[idxs_unlabeled]
        selected_labels = pool_labels[idxs_unlabeled]
        pool_data = np.delete(pool_data, idxs_unlabeled)
        pool_labels = np.delete(pool_labels, idxs_unlabeled)
        trainloader = DataLoader(Dataset(selected_data, selected_labels))
        criterion = nn.CrossEntropyLoss()
        optimizer = optim.Adam(net.parameters())
        for epoch in range(train_epochs):
            train(net, trainloader, criterion, optimizer)
        pool_loader = torch.utils.data.DataLoader(Dataset(pool_data, pool_labels))
        #Query unlabeled pool samples by using our calibrated uncertainty sampling.
        idxs_unlabeled = sampling(net, pool_loader, select_samples, trainloader)
        test_acc, test_ece = test_model(net, device, testloader)
```

## B.3 ADDITIONAL RESULTS

### B.3.1 DETAILED COMPARISON

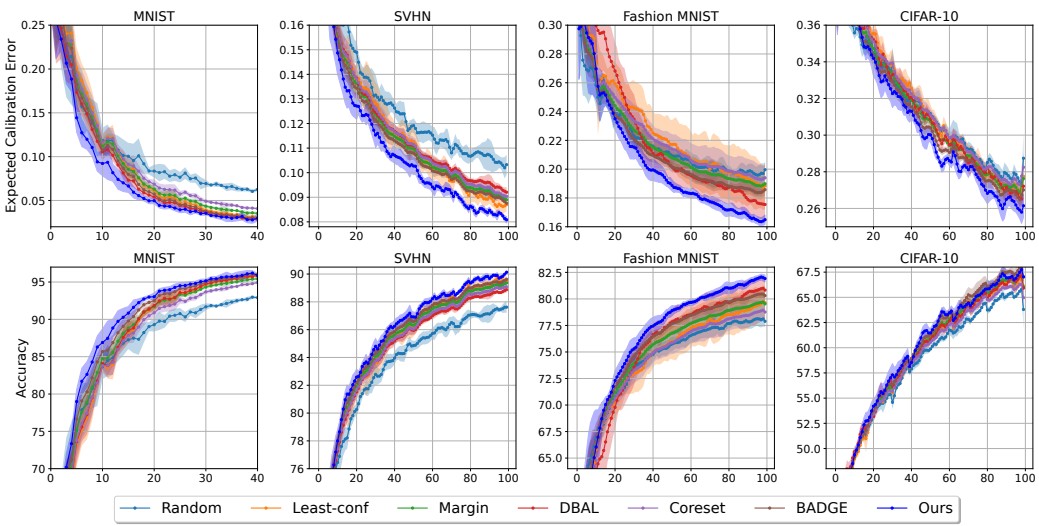

Figure 8: Calibration error (lower is better) and accuracy (higher is better) comparison with different baselines across query times $t \in [T]$ on the test set with different datasets and model architectures. Intervals for each line in the graph depict our results across 10 runs. Table details are in Tab. 1. Difference details are in Fig. 9.

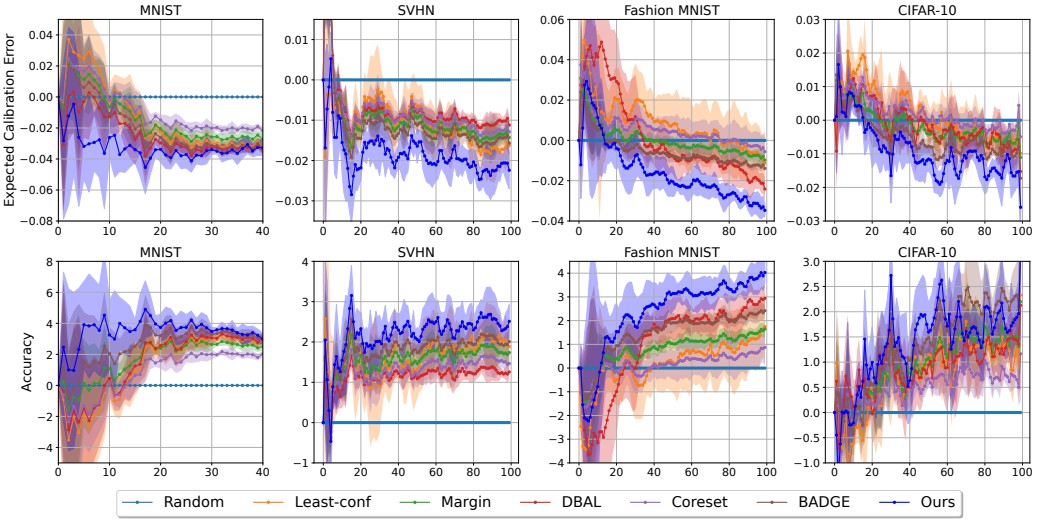

Figure 9: Plot the difference between a method and random selection, where each line represents the results value of the method minus the results value of the random selection baseline from Tab. 1 and Fig. 8. **Our method consistently outperforms other baselines at almost every time step** $t \in [T]$ **across different settings by lower ECE and higher accuracy.**

Table 2: Calibration error (lower is better) and accuracy (higher is better) comparison with different baselines across query times $t \in \{T/4, T/2, 3T/4, T\}$, where $T = 8$ on ImageNet. Scores are reported by mean $\pm$ standard deviation from 10 runs. Gray rows indicate our proposed method. Best scores with the significant test are marked in **bold**. Figure details are in Fig. 6. **Our algorithm consistently outperforms other AF baselines in this large-scale setting**. We additionally compare with the CAMPAL (i.e., CAMPAL$_{\text{BADGE}}^{\text{CHAMBER}}$) (Yang et al., 2023). **The combination of data-augmentation techniques with CAMPAL in training and our proposed AF in query time achieves the best performance.**

| Method | Expected Calibration Error ($\downarrow$) | | | | Accuracy ($\uparrow$) | | | |
|---|---|---|---|---|---|---|---|---|
| | $t = T/4$ | $t = T/2$ | $t = 3T/4$ | $t = T$ | $t = T/4$ | $t = T/2$ | $t = 3T/4$ | $t = T$ |
| Random | $0.3168 \pm 0.006$ | $0.2732 \pm 0.005$ | $0.2492 \pm 0.004$ | $0.2402 \pm 0.003$ | $55.68 \pm 0.3$ | $57.80 \pm 0.2$ | $59.11 \pm 0.2$ | $60.06 \pm 0.2$ |
| Least-conf | $0.2995 \pm 0.005$ | $0.2620 \pm 0.005$ | $0.2412 \pm 0.004$ | $0.2348 \pm 0.003$ | $55.86 \pm 0.2$ | $58.40 \pm 0.2$ | $59.78 \pm 0.2$ | $60.66 \pm 0.2$ |
| Rand-Entropy | $0.2988 \pm 0.006$ | $0.2613 \pm 0.005$ | $0.2406 \pm 0.004$ | $0.2341 \pm 0.003$ | $55.87 \pm 0.3$ | $58.44 \pm 0.2$ | $59.81 \pm 0.2$ | $60.68 \pm 0.2$ |
| Margin | $0.2969 \pm 0.005$ | $0.2599 \pm 0.004$ | $0.2379 \pm 0.003$ | $0.2297 \pm 0.002$ | $55.99 \pm 0.2$ | $58.60 \pm 0.2$ | $59.90 \pm 0.1$ | $60.68 \pm 0.1$ |
| BALD | $0.3018 \pm 0.006$ | $0.2628 \pm 0.005$ | $0.2397 \pm 0.004$ | $0.2315 \pm 0.003$ | $55.92 \pm 0.3$ | $58.42 \pm 0.2$ | $59.70 \pm 0.2$ | $60.53 \pm 0.2$ |
| BatchBALD | $0.3022 \pm 0.005$ | $0.2623 \pm 0.005$ | $0.2388 \pm 0.003$ | $0.2302 \pm 0.003$ | $55.95 \pm 0.3$ | $58.46 \pm 0.2$ | $59.79 \pm 0.2$ | $60.66 \pm 0.2$ |
| Coreset | $0.3082 \pm 0.005$ | $0.2676 \pm 0.004$ | $0.2452 \pm 0.003$ | $0.2375 \pm 0.003$ | $55.77 \pm 0.2$ | $58.10 \pm 0.2$ | $59.45 \pm 0.2$ | $60.36 \pm 0.2$ |
| Cluster-Margin | $0.2961 \pm 0.004$ | $0.2593 \pm 0.004$ | $0.2375 \pm 0.003$ | $0.2292 \pm 0.002$ | $55.99 \pm 0.2$ | $58.67 \pm 0.2$ | $59.97 \pm 0.2$ | $60.70 \pm 0.1$ |
| BADGE | $0.2920 \pm 0.004$ | $0.2570 \pm 0.004$ | $0.2361 \pm 0.003$ | $0.2279 \pm 0.002$ | $56.06 \pm 0.2$ | $58.77 \pm 0.2$ | $60.10 \pm 0.1$ | $60.84 \pm 0.1$ |
| Ours | $\mathbf{0.2868} \pm 0.004$ | $\mathbf{0.2523} \pm 0.003$ | $\mathbf{0.2301} \pm 0.002$ | $\mathbf{0.2228} \pm 0.001$ | $\mathbf{56.16} \pm 0.2$ | $\mathbf{59.04} \pm 0.2$ | $\mathbf{60.28} \pm 0.1$ | $\mathbf{61.00} \pm 0.1$ |
| CAMPAL & BADGE | $0.2795 \pm 0.004$ | $0.2465 \pm 0.004$ | $0.2249 \pm 0.003$ | $0.2161 \pm 0.002$ | $57.96 \pm 0.2$ | $61.00 \pm 0.2$ | $62.42 \pm 0.1$ | $63.06 \pm 0.1$ |
| CAMPAL & Ours | $\mathbf{0.2623} \pm 0.004$ | $\mathbf{0.2332} \pm 0.003$ | $\mathbf{0.2101} \pm 0.002$ | $\mathbf{0.2011} \pm 0.001$ | $\mathbf{58.86} \pm 0.2$ | $\mathbf{61.86} \pm 0.2$ | $\mathbf{63.12} \pm 0.1$ | $\mathbf{63.65} \pm 0.1$ |

## B.3.2 EVALUATION ON AN IMBALANCED BENCHMARK

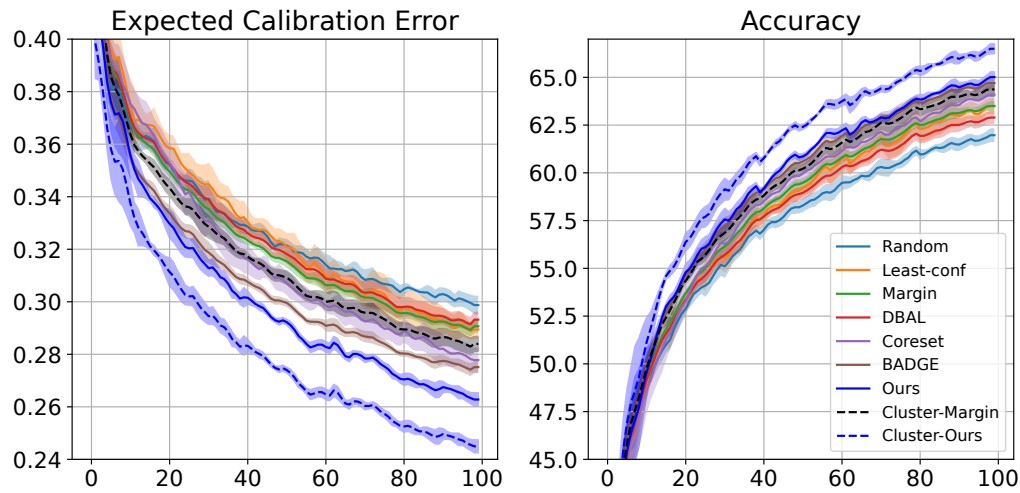

Figure 10: Calibration error (lower is better) and accuracy (higher is better) comparison with different baselines across query times $t \in [T]$, where $T = 100$ on an imbalanced setting with CIFAR-10-LT. Intervals for each line in the graph depict our results across 10 runs. Table details are in Tab. 3.

It is worth noticing that our AF aims to improve uncertainty-based sampling methods and not to focus on diversity. The reason is that we aim to improve not only the accuracy but also the uncertainty estimation quality of DNN in AL. This is crucial because it helps determine when an AI model's predictions can be trusted, especially in safety-critical applications.

That said, one factor that may impact the performance of uncertainty-based methods is the imbalanced class distributions of the dataset. Hence, we next test the robustness of our method on an imbalanced setting and how our method can extend to address the potential lack of diversity in the selected sample. In particular, we provide our results on Long-Tail CIFAR-10 (Cui et al., 2019) with an imbalance factor of 50 over ten runs. For each run, we initialize with 1000 randomly selected data points from the training dataset. Tab. 3 shows results with ResNet-18, data augmentation (Yang et al., 2023), the number of AL rounds $T = 100$, and the number of queried samples $k = 100$.

From Tab. 3 and Fig. 10, firstly, we observe that diversity-based methods (e.g., Coreset, BADGE) have a higher accuracy than uncertainty-based methods (e.g., Least-conf, Rand-Entropy, Margin). This is because the diversity-based method helps query balance class samples and uncalibrated uncertainty-based methods cause biased sampling. Notably, our method is more calibrated and can mitigate the biased sampling problem, leading to higher accuracy than other uncertainty-based baselines.

Secondly, compared to the best method, i.e., BADGE, although our method has only slightly higher accuracy by not focusing on diversity, it still outperforms BADGE in uncertainty estimation quality with a significantly lower ECE. This shows that our AF is still valuable in high-stakes applications, where humans need to know when an AI model's predictions can be trusted to make a final decision.

Finally, our method can address the potential lack of diversity in the selected samples by combining with other diversity-based approaches. For example, our AF can incorporate two-stage methods to become a hybrid AF (i.e., uncertainty-based & diversity-based) like Cluster-Margin. Specifically, we can extend our framework to Cluster-Ours by replacing Margin Step 8 in Alg.2 (Citovsky et al., 2021) with our proposed AF, i.e., select $k_m$ samples by our AF in Alg. 1. After that, we can follow the remaining steps in Alg.2 (Citovsky et al., 2021) to select $k_t$ from $k_m$ samples by a cluster-based approach to improve diversity. Tab. 3 and Fig. 10 show that Cluster-Ours can address the potential lack of diversity on this imbalanced dataset and can bring out the best performance with the highest accuracy and the lowest ECE. This once again confirms the benefits of our proposed AF and its potential to extend to improve trustworthiness across AL settings.

Table 3: Calibration error (lower is better) and accuracy (higher is better) comparison with different baselines across query times $t \in \{T/4, T/2, 3T/4, T\}$, where $T = 8$ on CIFAR-10-LT. Scores are reported by mean $\pm$ standard deviation from 10 runs. Gray rows indicate our proposed method. Best scores with the significant test are marked in **bold**. Figure details are in Fig. 10. **Our algorithm has competitive results in accuracy and outperforms other AF baselines in calibration**. We additionally extend our AF to a two-stage approach with Cluster-Ours to improve the diversity in the selected samples (i.e., select $k_m$ samples by our AF, then select $k_t$ from $k_m$ samples by a cluster-based approach to improve diversity (Citovsky et al., 2021).). **Cluster-Ours achieves the best performance and outperforms other two-stage-based AL** (e.g., Rand-Entropy, Cluster-Margin).

| Method | Expected Calibration Error (↓) | | | | Accuracy (↑) | | | |
|---|---|---|---|---|---|---|---|---|
| | $t = T/4$ | $t = T/2$ | $t = 3T/4$ | $t = T$ | $t = T/4$ | $t = T/2$ | $t = 3T/4$ | $t = T$ |
| Random | $0.346_{\pm 0.002}$ | $0.322_{\pm 0.002}$ | $0.307_{\pm 0.002}$ | $0.299_{\pm 0.003}$ | $53.9_{\pm 0.3}$ | $58.2_{\pm 0.3}$ | $60.4_{\pm 0.2}$ | $61.9_{\pm 0.3}$ |
| Least-conf | $0.351_{\pm 0.006}$ | $0.322_{\pm 0.005}$ | $0.304_{\pm 0.008}$ | $0.289_{\pm 0.004}$ | $54.5_{\pm 0.5}$ | $59.1_{\pm 0.5}$ | $61.6_{\pm 0.7}$ | $63.4_{\pm 0.4}$ |
| Margin | $0.343_{\pm 0.002}$ | $0.316_{\pm 0.002}$ | $0.299_{\pm 0.003}$ | $0.290_{\pm 0.002}$ | $54.8_{\pm 0.2}$ | $59.4_{\pm 0.3}$ | $61.8_{\pm 0.4}$ | $63.4_{\pm 0.2}$ |
| BALD | $0.346_{\pm 0.003}$ | $0.319_{\pm 0.003}$ | $0.303_{\pm 0.005}$ | $0.293_{\pm 0.003}$ | $54.4_{\pm 0.3}$ | $58.8_{\pm 0.4}$ | $61.3_{\pm 0.5}$ | $62.8_{\pm 0.3}$ |
| BatchBALD | $0.342_{\pm 0.003}$ | $0.312_{\pm 0.003}$ | $0.300_{\pm 0.006}$ | $0.285_{\pm 0.004}$ | $54.6_{\pm 0.5}$ | $59.1_{\pm 0.4}$ | $61.6_{\pm 0.5}$ | $63.1_{\pm 0.4}$ |
| Coreset | $0.341_{\pm 0.008}$ | $0.308_{\pm 0.005}$ | $0.291_{\pm 0.005}$ | $0.278_{\pm 0.005}$ | $54.9_{\pm 0.8}$ | $59.8_{\pm 0.4}$ | $62.3_{\pm 0.4}$ | $64.0_{\pm 0.5}$ |
| BADGE | $0.327_{\pm 0.003}$ | $0.300_{\pm 0.002}$ | $0.285_{\pm 0.002}$ | $0.275_{\pm 0.002}$ | $55.8_{\pm 0.4}$ | $60.5_{\pm 0.3}$ | $63.1_{\pm 0.4}$ | $64.6_{\pm 0.4}$ |
| Ours | $\mathbf{0.320}_{\pm 0.003}$ | $\mathbf{0.292}_{\pm 0.001}$ | $\mathbf{0.276}_{\pm 0.002}$ | $\mathbf{0.262}_{\pm 0.002}$ | $\mathbf{56.0}_{\pm 0.4}$ | $\mathbf{60.8}_{\pm 0.2}$ | $\mathbf{63.2}_{\pm 0.2}$ | $\mathbf{65.0}_{\pm 0.3}$ |
| Rand-Entropy | $0.343_{\pm 0.004}$ | $0.317_{\pm 0.003}$ | $0.296_{\pm 0.002}$ | $0.287_{\pm 0.002}$ | $54.9_{\pm 0.4}$ | $59.3_{\pm 0.3}$ | $61.9_{\pm 0.5}$ | $63.6_{\pm 0.5}$ |
| Cluster-Margin | $0.336_{\pm 0.003}$ | $0.310_{\pm 0.003}$ | $0.294_{\pm 0.002}$ | $0.284_{\pm 0.002}$ | $55.4_{\pm 0.3}$ | $60.1_{\pm 0.3}$ | $62.7_{\pm 0.5}$ | $64.3_{\pm 0.4}$ |
| Cluster-Ours | $\mathbf{0.302}_{\pm 0.003}$ | $\mathbf{0.274}_{\pm 0.001}$ | $\mathbf{0.258}_{\pm 0.001}$ | $\mathbf{0.244}_{\pm 0.002}$ | $\mathbf{57.6}_{\pm 0.4}$ | $\mathbf{62.4}_{\pm 0.1}$ | $\mathbf{64.7}_{\pm 0.1}$ | $\mathbf{66.5}_{\pm 0.2}$ |

### B.3.3 CALIBRATION ERROR ON THE UNLABELED POOL

To verify the benefits of our AF in terms of improving **uncertainty quantification on the unlabeled pool**, we additionally compare our proposed method with other baselines. We summarize this result in Fig. 11 and Tab. 4. Recall that for a fair comparison, we set the random seeds across baselines to be the same. As a result, for every round $t = 0$, the performance across methods is exactly similar. From these figures and the tables, we can also see that our method outperforms other baselines in every AL rounds across different settings by having the lowest ECE. For instance, its ECE is lower than others by more than 0.026 at $t = 10$ on MNIST, 0.01 at $t = 100$ on SVHN, 0.016 at $t = 75$ on F-MNIST, and 0.012 on CIFAR-10. It is also worth noticing that the ECE on the unlabeled pool in Fig. 11 is generally lower than on the unseen data in Fig. 8. This is because we can observe the samples of the unlabeled pool during our calibration estimation in Equation 9. Finally, to be summarized, from Fig. 8 and Fig. 11, we can see that the effectiveness of our proposed AF in improving model uncertainty estimation performance in both the unlabeled pool and the test set.

Hence, by this high-uncertainty estimation quality model, we can leverage calibrated uncertainty sampling to query informative samples to enhance generalization performance.

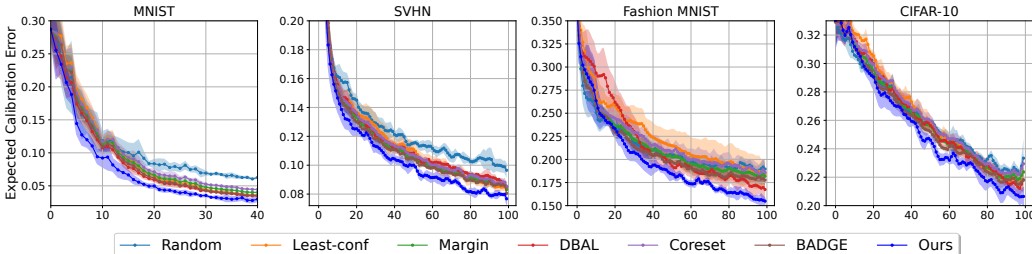

Figure 11: Calibration error on the unlabeled pool (lower is better) comparison with different baselines across query times $t \in [T]$ with different datasets and model architectures. Intervals for each line in the graph depict our results across 10 runs. Table details are in Tab. 4. **Our method achieves the lowest calibration error on the unlabeled pool across every** $t \in [T]$.

Table 4: Calibration error on the unlabeled pool (lower is better) comparison with different baselines across query times $t \in \{T/4, T/2, 3T/4, T\}$, where $T = 40$ on MNIST and $T = 100$ on other datasets. Scores are reported by mean $\pm$ standard deviation from 10 runs. Gray rows indicate our proposed method. Best scores with the significant test are marked in **bold**. Figure details are in Fig. 11.

| Dataset | Method | Expected Calibration Error | | | | |
|---|---|---|---|---|---|---|
| | | $t = 0$ | $t = T/4$ | $t = T/2$ | $t = 3T/4$ | $t = T$ |
| MNIST | Random | $0.287_{\pm 0.019}$ | $0.130_{\pm 0.031}$ | $0.084_{\pm 0.004}$ | $0.070_{\pm 0.007}$ | $0.060_{\pm 0.002}$ |
| | Least-conf | $0.287_{\pm 0.019}$ | $0.137_{\pm 0.013}$ | $0.068_{\pm 0.003}$ | $0.046_{\pm 0.003}$ | $0.037_{\pm 0.001}$ |
| | Margin | $0.287_{\pm 0.019}$ | $0.130_{\pm 0.014}$ | $0.069_{\pm 0.003}$ | $0.049_{\pm 0.003}$ | $0.040_{\pm 0.001}$ |
| | BALD | $0.287_{\pm 0.019}$ | $0.120_{\pm 0.013}$ | $0.061_{\pm 0.003}$ | $0.043_{\pm 0.002}$ | $0.035_{\pm 0.001}$ |
| | Coreset | $0.287_{\pm 0.019}$ | $0.135_{\pm 0.018}$ | $0.073_{\pm 0.003}$ | $0.054_{\pm 0.004}$ | $0.045_{\pm 0.001}$ |
| | BADGE | $0.287_{\pm 0.019}$ | $0.125_{\pm 0.010}$ | $0.064_{\pm 0.003}$ | $0.045_{\pm 0.002}$ | $0.036_{\pm 0.001}$ |
| | Ours | $\mathbf{0.287}_{\pm 0.019}$ | $\mathbf{0.094}_{\pm 0.011}$ | $\mathbf{0.050}_{\pm 0.003}$ | $\mathbf{0.035}_{\pm 0.002}$ | $\mathbf{0.028}_{\pm 0.004}$ |
| SVHN | Random | $0.437_{\pm 0.039}$ | $0.135_{\pm 0.005}$ | $0.116_{\pm 0.003}$ | $0.108_{\pm 0.005}$ | $0.097_{\pm 0.003}$ |
| | Least-conf | $0.437_{\pm 0.039}$ | $0.133_{\pm 0.009}$ | $0.111_{\pm 0.007}$ | $0.096_{\pm 0.004}$ | $0.085_{\pm 0.004}$ |
| | Margin | $0.437_{\pm 0.039}$ | $0.128_{\pm 0.004}$ | $0.106_{\pm 0.003}$ | $0.095_{\pm 0.003}$ | $0.083_{\pm 0.003}$ |
| | BALD | $0.437_{\pm 0.039}$ | $0.127_{\pm 0.004}$ | $0.107_{\pm 0.003}$ | $0.098_{\pm 0.004}$ | $0.087_{\pm 0.005}$ |
| | Coreset | $0.437_{\pm 0.039}$ | $0.131_{\pm 0.005}$ | $0.108_{\pm 0.003}$ | $0.097_{\pm 0.002}$ | $0.086_{\pm 0.004}$ |
| | BADGE | $0.437_{\pm 0.039}$ | $0.125_{\pm 0.002}$ | $0.105_{\pm 0.003}$ | $0.092_{\pm 0.003}$ | $0.084_{\pm 0.004}$ |
| | Ours | $\mathbf{0.437}_{\pm 0.039}$ | $\mathbf{0.120}_{\pm 0.003}$ | $\mathbf{0.101}_{\pm 0.004}$ | $\mathbf{0.085}_{\pm 0.002}$ | $\mathbf{0.074}_{\pm 0.003}$ |
| F-MNIST | Random | $0.366_{\pm 0.034}$ | $0.230_{\pm 0.009}$ | $0.204_{\pm 0.007}$ | $0.196_{\pm 0.010}$ | $0.194_{\pm 0.008}$ |
| | Least-conf | $0.366_{\pm 0.034}$ | $0.246_{\pm 0.017}$ | $0.215_{\pm 0.017}$ | $0.200_{\pm 0.021}$ | $0.185_{\pm 0.014}$ |
| | Margin | $0.366_{\pm 0.034}$ | $0.234_{\pm 0.007}$ | $0.204_{\pm 0.004}$ | $0.190_{\pm 0.009}$ | $0.182_{\pm 0.006}$ |
| | BALD | $0.366_{\pm 0.034}$ | $0.249_{\pm 0.020}$ | $0.198_{\pm 0.012}$ | $0.183_{\pm 0.013}$ | $0.169_{\pm 0.014}$ |
| | Coreset | $0.366_{\pm 0.034}$ | $0.238_{\pm 0.011}$ | $0.209_{\pm 0.012}$ | $0.196_{\pm 0.015}$ | $0.184_{\pm 0.009}$ |
| | BADGE | $0.366_{\pm 0.034}$ | $0.225_{\pm 0.004}$ | $0.198_{\pm 0.003}$ | $0.186_{\pm 0.006}$ | $0.180_{\pm 0.008}$ |
| | Ours | $\mathbf{0.366}_{\pm 0.034}$ | $\mathbf{0.220}_{\pm 0.010}$ | $\mathbf{0.185}_{\pm 0.006}$ | $\mathbf{0.167}_{\pm 0.005}$ | $\mathbf{0.155}_{\pm 0.003}$ |
| CIFAR-10 | Random | $0.329_{\pm 0.017}$ | $0.285_{\pm 0.004}$ | $0.260_{\pm 0.002}$ | $0.237_{\pm 0.003}$ | $0.234_{\pm 0.013}$ |
| | Least-conf | $0.329_{\pm 0.017}$ | $0.293_{\pm 0.007}$ | $0.260_{\pm 0.006}$ | $0.234_{\pm 0.008}$ | $0.224_{\pm 0.008}$ |
| | Margin | $0.329_{\pm 0.017}$ | $0.287_{\pm 0.003}$ | $0.259_{\pm 0.003}$ | $0.233_{\pm 0.003}$ | $0.224_{\pm 0.008}$ |
| | BALD | $0.329_{\pm 0.017}$ | $0.294_{\pm 0.010}$ | $0.278_{\pm 0.015}$ | $0.235_{\pm 0.003}$ | $0.218_{\pm 0.005}$ |
| | Coreset | $0.329_{\pm 0.017}$ | $0.290_{\pm 0.002}$ | $0.260_{\pm 0.003}$ | $0.235_{\pm 0.003}$ | $0.229_{\pm 0.010}$ |
| | BADGE | $0.329_{\pm 0.017}$ | $0.283_{\pm 0.005}$ | $0.252_{\pm 0.001}$ | $0.230_{\pm 0.002}$ | $0.219_{\pm 0.009}$ |
| | Ours | $\mathbf{0.329}_{\pm 0.017}$ | $\mathbf{0.278}_{\pm 0.009}$ | $\mathbf{0.247}_{\pm 0.004}$ | $\mathbf{0.227}_{\pm 0.003}$ | $\mathbf{0.206}_{\pm 0.006}$ |

### B.3.4 ADDITIONAL ABLATION STUDIES

Table 5: Frequency of samples selected based on calibration error versus uncertainty in our AF across query times $t \in \{T/4, T/2, 3T/4, T\}$, where $T = 100$, with the number of queried samples is $k = 50$ on SVHN. **The number of selected samples based on calibration error is high at early rounds. The model can then improve the calibration performance. As a result, in later rounds, the calibration error is lower and more uniform across samples (Fig. 3, Fig. 11), leading to a higher proportion of samples being selected by uncertainty sampling**.

| Time $t$ | $t = T/4$ | $t = T/2$ | $t = 3T/4$ | $t = T$ |
|---|---|---|---|---|
| # samples selected only based on calibration error | $48.0_{\pm 1.5}$ | $22.5_{\pm 3.2}$ | $17.2_{\pm 3.5}$ | $6.1_{\pm 2.8}$ |
| # samples selected additionally based on uncertainty | $2.0_{\pm 1.1}$ | $27.5_{\pm 3.6}$ | $32.8_{\pm 3.8}$ | $43.9_{\pm 2.0}$ |

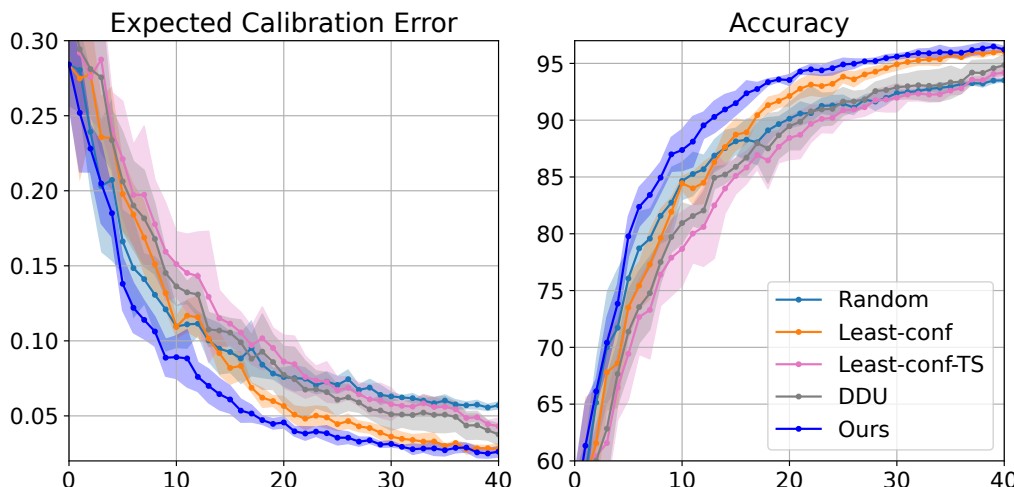

Figure 12: Calibration and accuracy comparison with the post-hoc calibrated uncertainty sampling baseline on MNIST. **To improve calibration performance, these post-hoc methods sacrifice samples in training, resulting in a significantly lower accuracy. Meanwhile, the calibration performance depends on model confidence and accuracy. As a result, their results are significantly worse than our AF**. We also compare with DDU and it is worth noting that DDU (Mukhoti et al., 2023) uses Temperature Scaling, while our setting does not use this post-hoc recalibration technique.

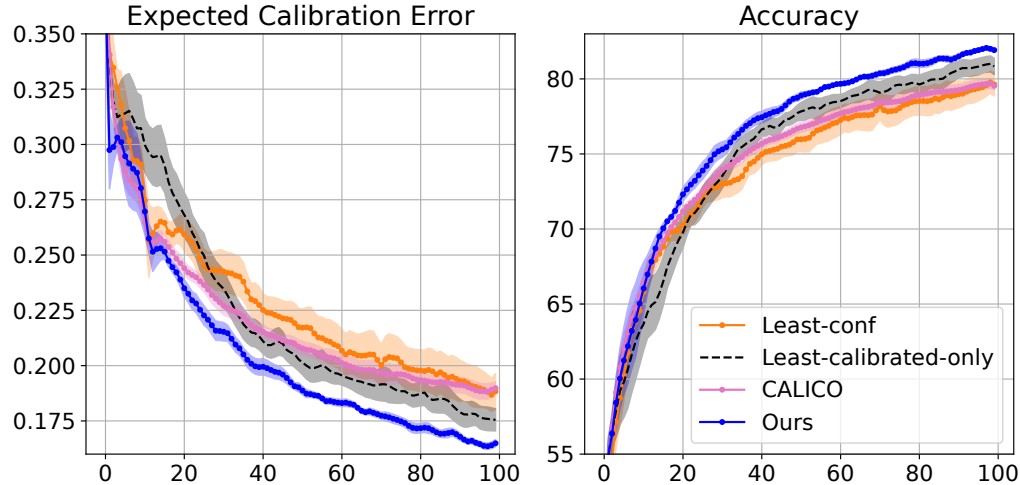

Figure 13: Calibration and accuracy comparison between query by only using the least calibrated samples, and ours AF on Fashion-MNIST. **Our AF results in a better performance with a lower ECE and higher accuracy, confirming the effectiveness of the lexicographic order during query time**. We also compare with CALICO and similar observations to Querol et al. (2024), the improvement of CALICO over Least-conf is minor (even worse in some settings), as the acquisition strategy stays the same as Least-conf and the joint training is in a semi-supervised way.

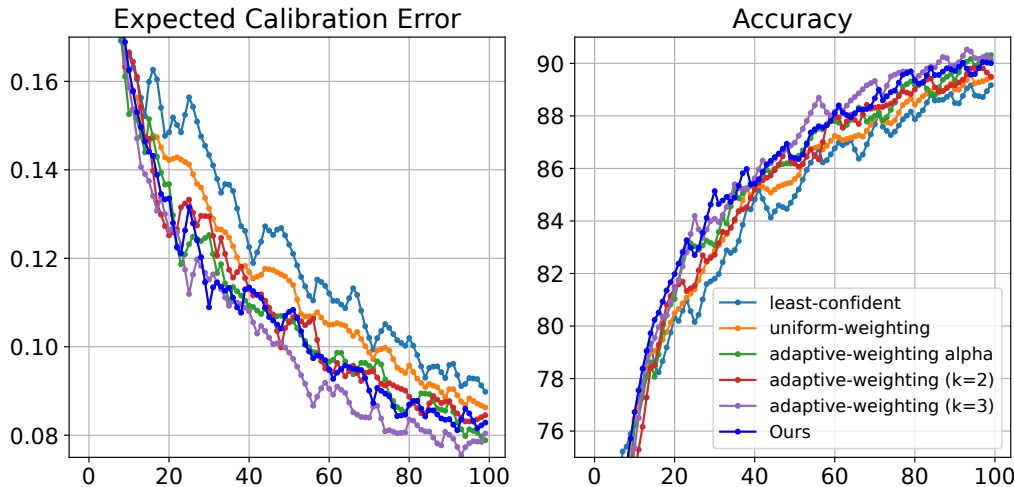

Figure 14: Our lexicographic order ablation study by comparing with **our proposed other weighting combination strategies**. The weighted combination ablation study on SVHN, including: (1) a uniform weighted combination, i.e., ranking by $(calibration + uncertainty)/2$; adaptive weighted combination, i.e., (2) ranking by $(\alpha * calibration + (1 - \alpha) * uncertainty)$, where $\alpha = $ mean(calibration error on the unlabeled pool) $\in [0, 1]$; (3) ranking by $(k * \alpha * calibration + uncertainty)/(k * \alpha + 1)$, where $k = \{2, 3\}$. **With an appropriate weighting hyperparameter (e.g., $k = 3$), the adaptive weighting can further improve the performance. However, with inappropriate weighting (e.g., uniform weighting), it can lead to a degradation of performance. This suggests our lexicographic order is a more flexible strategy because it does not depend on $\alpha$ while still bringing out a good performance in practice.**

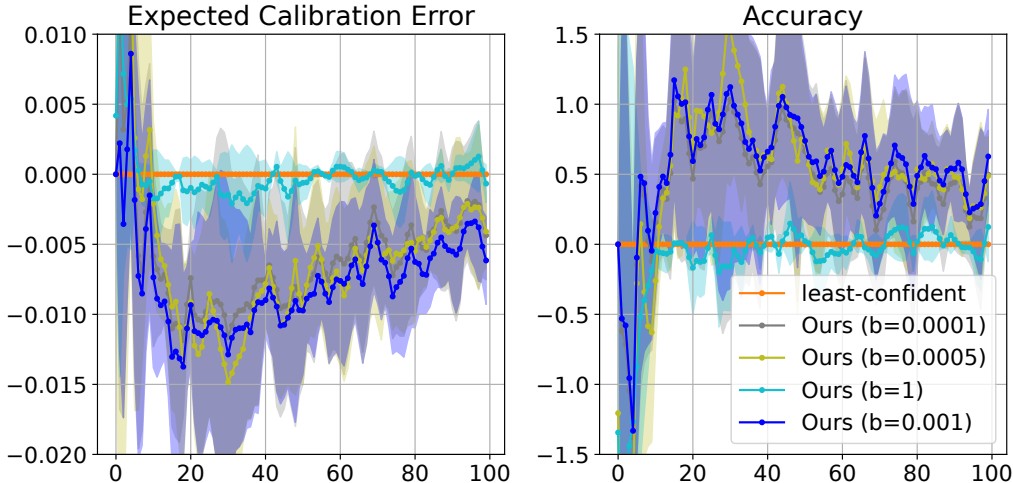

Figure 15: The choice of kernel bandwidth $b$, where each line represents the results value of the method minus the results value of the least-confident baseline on SVHN. Following (Popordanoska et al., 2022), we compare our kernel bandwidth $b = \{0.0001, 0.0005, 0.001, 1\}$. **We observe that there is no significant difference between our performances. Yet, if the bandwidth is too large, e.g., $b = 1$, it can lead to an oversmoothing problem. And, our ranking in calibration error becomes uniform, causing similar results with the least-confident baseline.**

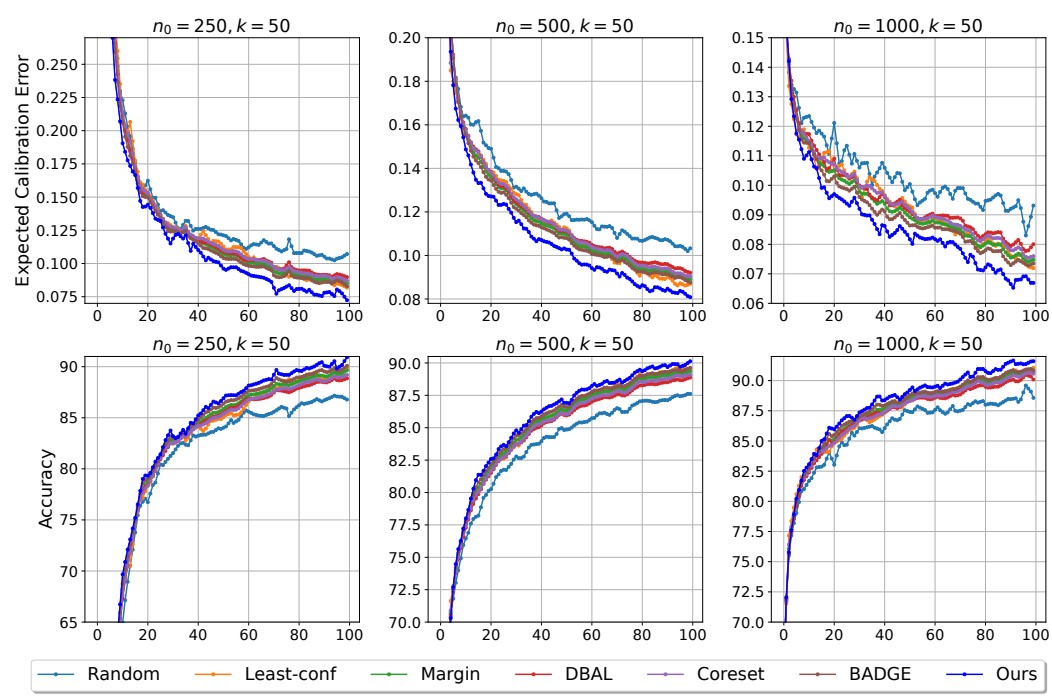

Figure 16: Calibration error (lower is better) and accuracy (higher is better) comparison across query times $t \in [T]$ on the SVHN with different numbers of warm-up samples $n_0 = \{250, 500, 1000\}$. **Our results are consistent across different numbers of warm-up samples.**

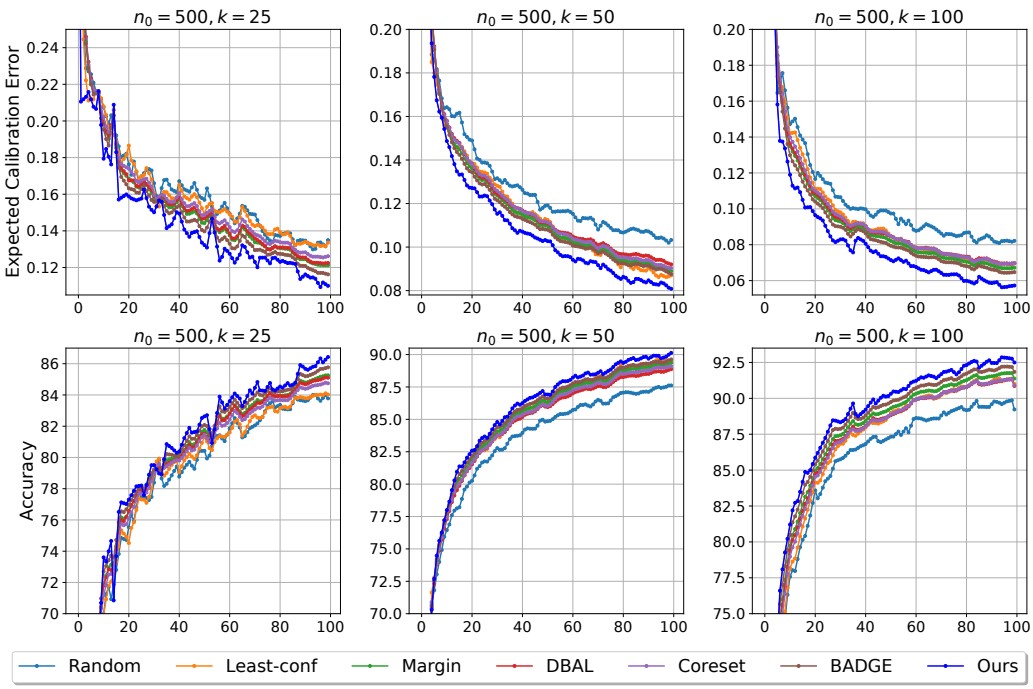

Figure 17: Calibration error (lower is better) and accuracy (higher is better) comparison across query times $t \in [T]$ on the SVHN with different numbers of queried samples $k = \{25, 50, 100\}$. **Our results are consistent across different numbers of queried samples.**

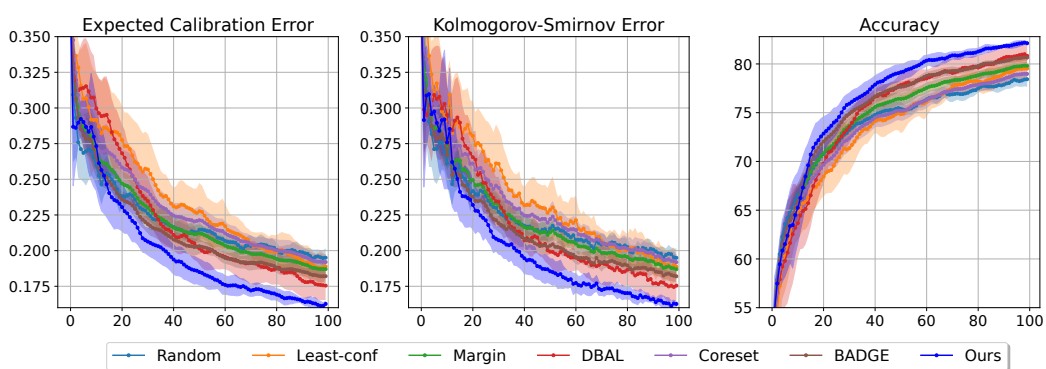

Figure 18: Expected calibration error, Kolmogorov-Smirnov (KS) error with top-1 prediction (Gupta et al., 2021) (lower is better), and accuracy (higher is better) comparison with different baselines across query times $t \in [T]$, where $T = 100$ on Fashion-MNIST. Intervals for each line in the graph depict our results across 10 runs. **We observe that the calibration error across ECE and KS metrics is consistent, and our method outperforms other baselines by lower ECE, KS error, and higher accuracy.**

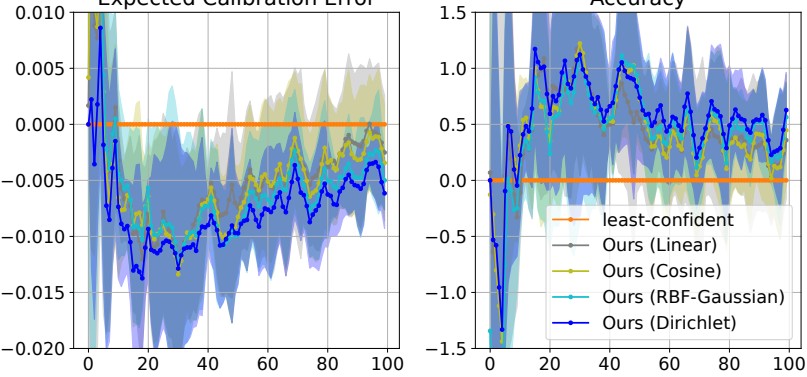

Figure 19: The choice of kernel, where each line represents the results value of the method minus the results value of the least-confident baseline on SVHN. We compare our default kernel (Dirichlet) with RBF-Gaussian, Cosine, and Linear. **We observe that the performance differences across kernels are relatively small. That said, Dirichlet and RBF-Gaussian are still slightly better than Cosine and Linear with lower calibration and higher accuracy.**

