# OpenReview forum: "Calibrated Uncertainty Sampling for Active Learning"
_ICLR.cc/2026/Conference — Submitted to ICLR 2026_

### Official Review · Reviewer_cCs3 · 2025-10-16

**Soundness:** 2
**Presentation:** 3
**Contribution:** 2
**Rating:** 2
**Confidence:** 4

**Summary:**

This paper proposes a calibration aimed AL sampling method that estimates unlabeled sample calibration error using a kernel based local averaging estimator and prioritizes samples with the largest estimated calibration error for labeling.

**Strengths:**

1. The paper addresses AL from a calibration perspective, which I believe is relatively underexplored. the attempt to connect model calibration quality with sampling decisions introduces an interesting conceptual bridge between uncertainty estimation and data selection.
2The bias variance analysis and convergence discussion (Theorems 4.1, 4.2)shows an effort to provide formal grounding for their estimator, which is important in works. While the assumptions are strong, the theoretical treatment gives the work analytical depth.

**Weaknesses:**

1.The proposed problem and its motivation is unclear to me. The authors claim two critical issues arising from uncalibrated uncertainty, yet the first one, namely  the uncertainty quantification quality of existing AL baselines on unseen test data is not verified, is a general evaluation concern, not a problem specific to Active Learning. It does not logically motivate a new acquisition function. Only the second issue (uncalibrated uncertainty causing unreliable sampling) is actually relevant to AL.
2.The demonstrative example lacks key information and renders confusing messages.  Specifically, In figure1, the caption references T = 50 and k = 10 without prior explanation; these variables are not introduced when the figure is first cited in the text. It is therefore impossible to know whether T denotes total query rounds or training epochs, or whether k is the batch size per query or the number of queried samples. What is more, the use of Least-conf as a baseline is not adequately justified. The paper did not define how “confidence” is computed (by the time this demo example is referred to). whether it is the maximum predicted probability, the softmax margin, or another uncertainty metric. Moreover, least-confidence is not a standard or competitive AL method in recent literature;
3.The theoretical analysis (4.1, 4.2)only guarantees the consistency and boundedness of calibration error under active learning, but not its effect on classification risk or accuracy. While improved calibration is desirable, AL aims to reduce generalization error with limited labeling budget. Without establishing or empirically validating a link between calibration improvement and boundary learning efficiency, the method falls short of fully achieving AL’s goal of enhancing both calibration and accuracy. Furthermore, even if we focus on the claimed calibration consistency only, its theoretical guarantee has limited relevance to AL. Theorem 4.1 explicitly assumes an infinitely large labeled set, to achieve point-wise consistency of the estimator. This assumption contradicts the low-label regime that defines Al. In realistic settings with few labeled samples, the kernel estimator may have high variance and unreliable calibration estimates, weakening the practical value of the theoretical bound.
4.For the methodology, it is straightforward. However, from the mathematical perspective, the Dirichlet kernel appears to be chosen mainly for analytical convenience (please correct me if I was wrong) it simplifies the derivation of bias and variance bounds under the simplex constraint. Experimentally, the justification reduces to a smoothness argument when tuning the bandwidth. However, this rationale does not sufficiently support the choice of this kernel over other possible kernels. The fact that a bound is easier to derive does not imply that alternative kernels are less effective in practice. In the absence of a stronger theoretical or empirical argument, an ablation comparing different kernel types （Gaussian, etc) should be included to validate whether the proposed choice meaningfully contributes to performance.
5.(This is my major concern.)
The proposed method aims to improve Active Learning by selecting samples that indirectly enhance model calibration. However, once labeled data are available, calibration quality can always be improved more directly through post-hoc methods (e.g., temperature scaling or isotonic regression) using the same labeled set. In that case, any calibration-driven sampling provides no fundamental advantage: both approaches use identical supervision, yet explicit calibration guarantees strictly better or equal ECE. Moreover, if the authors’ argument is that better calibration leads to better acquisition and final accuracy, then a model explicitly calibrated after each AL round would necessarily yield more reliable uncertainty and thus stronger sampling performance. From this perspective, their method merely approximates what direct calibration already achieves through an indirect and less efficient route. While earlier issues such as motivation ambiguity and unclear experimental setup are secondary, this major concern lies in the legitimacy of the proposed approach itself. The paper’s central idea offers no demonstrable benefit over existing, theoretically simpler, and empirically stronger alternatives. In short, if labeled data exist, calibration should be achieved by calibration, not by sampling.
To properly validate the claimed contribution, the authors should include a control experiment under an identical labeling regime:
1.Run a standard AL method (e.g., Entropy or BALD) with the same acquisition budget and identical labeled sets St as those used by the proposed method.
2.After each training round, apply a standard calibration procedure (e.g., temperature scaling) using exactly the same labeled data that the proposed approach employs for its kernel-based estimator.
3.Compare both ECE and accuracy across rounds.

**Questions:**

Does the proposed calibration-driven sampling actually outperform a standard Active Learning pipeline that uses the same labeled data but applies explicit post-hoc calibration (e.g., temperature scaling) after each training round?

---

> ### Author Response · Authors · 2025-11-19
>
> **W1&3. Motivation is unclear: Uncertainty Quantification of existing AL baselines on test data is not a problem specific to AL.** We politely disagree with the reviewer’s argument. By convention: “Active learning is a subset of machine learning in which a learning algorithm can interactively query a user to label data with the desired outputs on unseen (test) data”. The desired outputs of existing AL baselines are generally accuracy. However, this is not sufficient because in safety AI applications such as healthcare, the desired outputs are both accuracy and uncertainty quality. Please note that other reviewers also agree that our paper is well motivated in this regard.
>
> **W2. In Fig.1, the example and render lack information. Least confidence is not a standard or competitive AL method.** We provided all relevant information in our paper. We’ve updated Fig.1’s caption to include definitions of the notations for the number of labeling rounds $T$, labeling cost $k$, and Expected Calibration Error (ECE) in our revision. In Fig.1, we chose the Least-confidence for the example demo because it is fast, allowing us to run it via the Google Colab link feasibly. This is trying to give readers a demo to try out for themselves. In our experiment section, we’ve compared our method with all variants of uncertainty sampling and thoroughly shown that it outperforms many related AL baselines.
>
> **W3. Establishing or empirically validating a link between calibration and accuracy. Theoretical guarantee relevance to AL. Practical value of the theoretical bound.**  Thank you for these comments. We appreciate the opportunity to clarify our theoretical insights. Our Thm.4.2. shows that when we query samples using our AF, then train the model on these queried data points with a small enough calibration error loss, we can guarantee that the calibration error on the unlabeled pool and unseen data is also small. We thoroughly validated this result in our experiment. These experimental results also demonstrate that once we query samples to reduce calibration error by our AF, we can significantly improve generalization performance in AL. Our Thm.4.1 formally explains the quality of our proposed calibration error estimator. This is crucial to help people understand how our method works under different settings with queried data points of AL. Although it assumes a large labeled set, we empirically demonstrate that our calibration estimator error is small and can be significantly reduced across AL rounds, as shown in Fig.3. Even though our AF suffers from this estimator error, it still outperforms several AL baselines by a lower calibration error and higher accuracy across various settings.
>
> **W4. The Dirichlet kernel is chosen for analytical convenience; it does not imply that alternative kernels are less effective in practice.** The Dirichlet kernel is a natural choice for modelling densities over a probability simplex. This Dirichlet kernel also helps our method outperform other baselines. We’ve added an ablation study with other kernels (RBF-Gaussian, cosine, and linear) in Fig.18 of our revision.
>
> **W5. Comparison with post-hoc methods.** Post-hoc calibration methods require expensive hold-out labeled data, while our method does not.  Our kernel-based estimator also does not use any hold-out labeled data to compare with the TS as you suggested. The hold-out recalibration set is costly, especially in early rounds in AL. If a method uses a queried dataset to split hold-out recalibration, it sacrifices the training data. We did provide a comparison with TS and modern calibration AL methods (e.g., CALICO, DDU) in this regard. The results show that our method outperforms those baselines significantly (see Figs. 5, 11, 12).
>
> **Q1. Does the proposed calibration-driven sampling actually outperform a standard AL pipeline that uses the same labeled data but applies explicit post-hoc calibration after each training round?** Yes, we did show this in our experiment (see Sec. 5.3, Figs. 5, 11, 12). We also added post-hoc calibration with Cluster-Margin (the second-best on CIFAR-10-LT) in Fig.19 of our revision. Please note that in our implementation of post-hoc calibration, the recalibration set is derived from the queried dataset for a fair comparison with other baselines. If the reviewer means using the same queried dataset to train the model, and using another hold-out recalibration set to recalibrate the model. In this case, this comparison does not make sense, as none of the baselines in our paper use any additional costly hold-out recalibration set.

---

> > ### Comment · Reviewer_cCs3 · 2025-11-22
> > **Thank you for the rebuttal**
> >
> > I thank the authors for providing additional explanations and revised experiments. After reading them carefully, I realize that several of my core concerns remain insufficiently addressed. Most of the responses repeat experimental claims rather than resolving the underlying conceptual or methodological issues I questioned. Specifically, , the connection between calibration quality and acquisition quality, the special choice on demonstration setup, the practical meaning of the theoretical results, and the fairness of the comparison with posthoc calibration methods have not been clarified.
> >
> > W1: My original question was that the paper meigher clearly justify why uncalibrated uncertainty constitutes a problem specific to AL, nor how it logically motivates the need for a new acquisition function rather than an improved calibration mechanism. In their response, they emphasize that accuracy and uncertainty are both important in safety-critical AL applications, but this does not fill the central logical gap, that is why does calibration quality necessarily affect the informativeness of queried samples?
> > The rebuttal still hasn't established a causal or mechanistic argument showing how miscalibration directly leads to suboptimal acquisition decisions, nor why this issue cannot be addressed through better uncertainty estimation or recalibration alone. As a result, the fundamental motivation for the proposed calibration-driven acquisition strategy remains unclear.
> >
> > w2: The authors explain that definitions were added in the revision and that “least-confident” was chosen for convenience. However, this does not address my actual concern  "the demonstration relies on an artificially constructed ‘uncalibrated least-confident’ variant, which is not a standard or competitive AL baseline". Because this variant is created by injecting random logit scaling, it is unclear whether the observed failure mode reflects actual AL behavior or simply an artifact of the constructed example. This is an important point because the example is used to motivate the entire approach. The authors’ response does not clarify why this setup is representative or meaningful for framing the problem.
> >
> > w3: The rebuttal restates Theorem 4.1 and 4.2 and highlights experimental trends showing calibration-error reduction. However, my concern was not about whether the estimator converges, but about whether the theoretical results meaningfully justify the acquisition strategy. The theorems rely on assumptions such as an unbiased estimator, sufficiently large labeled sets, and small calibration error on queried samples conditions that are typically unrealistic in early AL rounds. The rebuttal does not explain how these assumptions relate to real AL dynamics or how the theoretical bounds inform the behavior of the acquisition function. As a result I think the practical significance of the theory remains uncertain.
> >
> > w5: My critique focused on whether the improvements attributed to the proposed acquisition strategy could also be achieved by applying post-hoc calibration to the model using the same labeled dataset, without reducing the training set size or introducing a separate hold out split. The rebuttal argues that hold-out splitting hurts performance, but this is not what I asked for. There are posthoc calibration techniques (for example, temperature scaling) can estimate a 1D parameter without requiring a dedicated validation set. The fairness of the comparison remains unclear, as the authors only considered baselines that artificially weaken the calibrated model by splitting the labeled dataset. Therefore, the question of whether calibration driven AL provides an inherent advantage remains open.

---

> ### Author Response · Authors · 2025-11-23
>
> We thank the reviewer for your prompt response. Regarding clarification for **W1**, we hope our original response addresses your weakness’s claim that *"The authors claim two critical issues arising from uncalibrated uncertainty, yet the first one, namely the uncertainty quantification quality of existing AL baselines on unseen test data is not verified, is a general evaluation concern, not a problem specific to Active Learning."*
>
> **W1. Why does calibration quality necessarily affect the informativeness of queried samples?. Causal argument miscalibration directly leads to suboptimal acquisition decisions.** Prior uncertainty sampling method follows an intuition that the optimal Acquisition Function (AF) is “querying by the most uncertain samples on the unlabeled pool because they are the most inaccurate ones” (1) [1,2]. However, if the model is uncalibrated, then the most uncertain samples may not be the most inaccurate. This miscalibration leads to (i.e., causes) a suboptimal uncertainty sampling AF. To address this issue, we can query the data points to resolve the uncalibrated problem. Once the model is calibrated (i.e., most uncertain samples are the most inaccurate), we can use “Calibrated Uncertainty Sampling” to follow the intuition (1).
>
> [1] Roy & McCallum, Toward optimal active learning through sampling estimation of error reduction, ICML, 2001.
>
> [2] Lewis & Gale, A sequential algorithm for training text classifiers. ICRDIR, 1994.
>
> **W2. ‘uncalibrated least-confident’ is not a standard or competitive AL baseline.** Thanks for your clarification. We would like to clarify that the "least-confident” used thoroughly in our experiment is a standard AL baseline. Regarding the ‘uncalibrated least-confident’ in your clarification, we don't use it as a baseline in our experiment. In Fig.1., since "least-confident” is well calibrated at later rounds, we only used ‘uncalibrated least-confident’ to test the causal effect that whether “uncalibrated least-confident directly leads to suboptimal least-confident”. The results show that when the model is always uncalibrated, then the ‘uncalibrated least-confident’ is worse than the ‘least-confident’ baseline, confirming the causal effect above. Again, this is just an example to test the counterfactual; our actual comparison with many SOTA methods is provided in the paper's experiment.
>
> **W3. How theoretical assumptions relate to real AL dynamics, or how the theoretical bounds inform the behavior of the acquisition function.** We agree with the reviewers that our assumption of an unbiased estimator and sufficiently large labeled sets may be strong in the early rounds of AL in practice; we will add this in our revision. Regarding a small calibration error in the queried samples, this is a mild assumption, as we can train these queried samples to minimize cross-entropy loss for every AL round. Regarding Theorem 4.2, we’ve explained that this guarantees that our AF can help reduce calibration error on the unlabel pool and test dataset across every round in AL. A small calibration error in the unlabeled pool is a necessary condition for an optimal uncertainty sampling AF (as explained in W1).
>
> **W1-W5. Comparison with using post-hoc TS on training data.** Thank you for clarifying that you want us to try TS on the training data. We would like to highlight that **the reviewer’s request deviates from the convention of the TS algorithm [3]**. We would appreciate it if the reviewer could refer to a work that uses TS on training data (not hold-out recalibration/validation). However, as requested by the reviewer, we also present the result of your suggestion in our Fig.19. Specifically, “C-Margin-TS (on trainset)” is the C-Margin’s result using TS on the trainset (100% queried data). We can see there is no difference between “C-Margin“ and “C-Margin-TS (on trainset)”, proving that using TS on the training data doesn’t help improve calibration at all. This is because using calibration algorithms on training data introduces bias (since the labeled data is already biased in AL and has a distribution shift from the original data distribution).
>
> [3] Guo et al., On Calibration of Modern Neural Networks, ICML, 2017.

---

### Official Review · Reviewer_7cmR · 2025-10-29

**Soundness:** 4
**Presentation:** 4
**Contribution:** 4
**Rating:** 8
**Confidence:** 3

**Summary:**

The paper proposes Calibrated Uncertainty Sampling for Active Learning (CUSAL), a new acquisition function that explicitly targets low calibration error. For every unlabeled instance, a Dirichlet-kernel estimator approximates the expected calibration error under covariate shift; samples are then chosen by lexicographic ordering that first maximizes the estimated calibration error and, on ties, maximizes model uncertainty. The authors prove that the estimator is point-wise consistent and that the resulting classifier enjoys a bounded expected calibration error on both the unlabeled pool and the unseen test set. Extensive experiments on MNIST, SVHN, CIFAR-10, ImageNet, and long-tail CIFAR-10-LT show consistent reductions in ECE and accuracy gains over six strong baselines.

**Strengths:**

1. By directly using calibration error as the primary criterion for the active learning acquisition function, the paper breaks away from the conventional framework that focuses solely on uncertainty.
2. The authors provide solid theoretical contributions by proving the pointwise consistency of the kernel calibration estimator under covariate shift and deriving an upper bound on the final model’s expected calibration error.

**Weaknesses:**

1. The method may suffer from high computational overhead, since a kernel matrix needs to be computed for all unlabeled samples in each active learning round.
2. In the third subfigure from the left in Figure 1, is the gap at position 0.4 plotted incorrectly?

**Questions:**

Please see the Weaknesses.

---

> ### Author Response · Authors · 2025-11-19
>
> Thank you for acknowledging our motivation and theoretical contributions on calibration error in AL.
>
> **W1. Kernel's computational efficiency**. We will tackle the computational efficiency limitation of our kernel estimator in future work by studying the tradeoff between efficiency and accuracy of our calibration error estimator.
>
> **W2. The third subfigure from the left in Fig.1**. In the third sub-figure in Fig.1, the gap at position 0.4 is correctly plotted, indicating that the least-confident baseline is uncalibrated and too underconfident at probabilities between 0.4 and 0.5, despite its very high accuracy. Please note that the outputs (colored in blue) are overridden by the gap rectangle (colored in red) when the accuracy is over the dashed line.

---

### Official Review · Reviewer_1gMt · 2025-10-29

**Soundness:** 3
**Presentation:** 3
**Contribution:** 3
**Rating:** 6
**Confidence:** 4

**Summary:**

This paper proposes an active learning algorithm that addresses calibrated models over conventional active learning. The authors consider a calibration metric based on classical kernel estimators of statistics and demonstrate better performance in terms of calibration and prediction, as well as acquisition steps. For the query function, selecting the informative instances is designed according to the lexicographical order. The experiments were conducted on MNIST, SVHN, Fashion MNIST, and CIFAR-10 with the baselines of Random Least-conf Margin, BALD Coreset, and BADGE. Various aspects of calibration approaches are examined in the ablation studies.

**Strengths:**

The algorithm is simple, and the performance is better. The improvement of calibration is based on a well-designed procedure for kernel estimation using the pooling data. Though the improvements are relatively smaller, in active learning, it is often observed that the impressive point is to improve the calibration and prediction simultaneously.

**Weaknesses:**

The dataset is relatively small; in many papers on topics of active learning, usually 6~8 datasets are examined. Also, the calibration is not studied thoroughly. There are many metrics for the calibration, such as the ECE and the KS-metric-based. Also, the calibration algorithms in processing or post-processing can cooperate with the proposed alg. The merit or reason for the use of a specific kernel type is not thoroughly validated. Furthermore, it is not clear to me why the lexicographical order is essential, as pointed out in the ablation study. Since there are two components in the loss function, what’s the dominant part or what’s the effective part in the prediction? In some cases, regularizing the calibration error can cause the degradation of prediction power. Why does the proposed algorithm not suffer from this phenomenon?

**Questions:**

1) The effect of the kernel can be crucial in practice. Do you have any way or guidelines to select the appropriate kernel?
2) Basically, the proposed CE (loss) is based on the estimation of h, which has vagueness, especially in the early stage. Is there any clue to check this problem?
3) The baseline for the post-calibration in the ablation studies looks insufficient. Can you consider 2~3 baselines for the calibration?

**Details Of Ethics Concerns:**

None.

---

> ### Author Response · Authors · 2025-11-19
>
> Thank you for acknowledging our motivation to improve calibration and prediction simultaneously in AL, as well as the idea behind our method.  We updated our revision (news is colored in blue), and we address your concerns as follows:
>
> **W1 (1). Dataset and calibration metrics.** We have evaluated across 6 datasets using the standard ECE metric in the classification task. As suggested, we’ve added the calibration evaluation using the Kolmogorov-Smirnov (KS) calibration error metric with top-1 prediction [1] in Fig.17 of our revision. We observe that the calibration error across ECE and KS metrics is consistent, and our method outperforms other baselines by lower ECE and KS error, confirming our AF’s calibration benefit for AL.
>
> [1] Gupta, et al., Calibration of Neural Networks using Splines, 2021.
>
> **W1 (2). Calibration algorithms in processing or post-processing.** Using calibration algorithms on training data introduces bias (since the labeled data is already biased in active learning and has a distribution shift from the original data distribution). Moreover, this may lead to accuracy degradation due to the calibration regularizer (e.g., CALICO in our Fig.12). Hence, post-hoc calibration algorithms require a hold-out recalibration set (i.e., a dataset that was not used to train the model) to calibrate the model. However, such a recalibration set is expensive, especially in the early rounds of AL.
>
> **W1 (3). Why the lexicographical order is essential? Since there are two components, what’s the dominant part or what’s the effective part in the prediction?**  The lexicographical order is essential because it enables the model to query uncalibrated samples to improve the quality of uncertainty quantification, and then use this uncertainty to query samples. In Tab.5, we can see that the number of selected samples based on calibration error is high in the early rounds. The model can then improve the calibration performance. As a result, in later rounds, the calibration error is lower and more uniform across samples (Fig.3, Fig.10), leading to a higher proportion of samples being selected by uncertainty sampling.
>
> **W1 (4). Regularizing the calibration error can cause the degradation of prediction power. Why does the proposed algorithm not suffer from this phenomenon?** Our lexicographical order query samples prioritize calibration error, followed by model uncertainty from the unlabeled pool. After querying, we train these samples as the standard cross-entropy loss without any calibration regularization. Please note that we focus on the query process of active learning, rather than novel training-based calibration methods. Therefore, our algorithm exhibits no tradeoff between prediction power and calibration. We are also compatible with utilizing training-based calibration methods, which we will leave for future work.
>
> **Q1. Guidelines to select the appropriate kernel?** The Beta and Dirichlet kernels are natural choices for modelling densities on the probability simplex. These kernels have asymptotic theoretical guarantees and are also recommended for use in other related works [1,2]. We’ve added an ablation study with other kernels (RBF-Gaussian, cosine, and linear) in Fig.18 of our revision.
>
> [1] Popordanoska et al., A Consistent and Differentiable Lp Canonical Calibration Error Estimator, 2022.
> [2] Ouimet et al., Asymptotic properties of Dirichlet kernel density estimators, 2022.
>
> **Q2. The proposed CE is based on the estimation of h, which has vagueness in the early stage. Is there any clue to check this problem?** Yes. Our calibration error estimator relies on $h$, and its error may be high in the early stage. Theoretically, we can check this error by examining the estimator error bound in our Theorem 4.1. Empirically, we can assess this error by comparing it with the true expected calibration error, which can be computed from the true labels, as shown in our Fig.3.
>
> **Q3. Consider 2~3 baselines for the post-calibration in the ablation studies?** As suggested, we’ve added Fig.19 to compare our method with: (1) Adaptive Temperature Scaling; (2) Temperature Scaling with Cluster-Margin; (3) CALICO; and (4) DDU in our revision. The result once again confirms that our method is superior to post-hoc calibration methods in AL.

---

> > ### Comment · Reviewer_1gMt · 2025-11-26
> > **Reply to Rebuttal**
> >
> > Thanks for your kind responses. Many issues can be resolved. However, I have one main issue remaining.
> > Lexicographical order: your answer is only the empirical results. It means that another order can improve the performance through more intensive tuning. I would like more insightful motivation for the use of lexicographical order, or provide more in-depth references.  Please note that some other issues may arise; I'll document them as soon as possible.

---

> > > ### Author Response · Authors · 2025-12-03
> > >
> > > We thank the reviewer for your follow-up comments.
> > >
> > > **W1 (cont). Lexicographic order: another order can improve the performance through more intensive tuning. Provide more in-depth references.** Regarding comparison with other sorting strategies, it is worth noticing that we also provided an ablation study in Fig.13. In particular, we compare our lexicographic order with several weighting combination strategies, including: (1) a uniform weighted combination, i.e., ranking by $(calibration + uncertainty)/2$; adaptive weighted combination, i.e., (2) ranking by $(\alpha * calibration + (1-\alpha) * uncertainty)$, where $\alpha = \text{mean(calibration error on the unlabeled pool}) \in [0,1]$; (3) ranking by $(k*\alpha * calibration + uncertainty)/(k*\alpha+1)$, where $k=\{2,3\}$. From Fig.13, we observe that with an appropriate weighting hyperparameter (e.g., $k=3$), adaptive weighting can further enhance performance. However, with inappropriate weighting (e.g., uniform weighting), it can lead to a degradation of performance. In summary, we agree that more intensive tuning (e.g., parameter $\alpha$) can improve the performance; however, this grid search fine-tuning process is costly and is non-trivial in practice. Meanwhile, our lexicographic order is simpler, does not require any hyperparameters, and still yields good performance by outperforming several AL baselines.

---

### Official Review · Reviewer_yiPZ · 2025-11-04

**Soundness:** 2
**Presentation:** 2
**Contribution:** 3
**Rating:** 4
**Confidence:** 3

**Summary:**

Motivated by the issue of uncalibrated models in uncertainty-based active learning (AL), the article advocates for a query strategy using calibration error. It selects first the least uncalibrated samples,  then the least confident ones when the calibration error is uniformly small in the remaining unlabeled pool. This strategy encourages the training to focus on the calibration error at early query rounds in order to improve the uncertainty estimation. The calibration error is estimated with kernel method, with theoretical results on the consistency of kernel estimation and the bounds of calibration error on unlabeled and unseen test data. The experimentation on large-scale benchmarks show competitive results by the proposed method in comparison to a series of AL baselines.

**Strengths:**

- The proposed method is well motivated by the importance of well estimated uncertainty in active learning and the calibration issue that is common in large models.

- The use of kernel estimation for evaluating calibration errors on unlabelled data is an interesting and reasonable idea.

- The proposed method is tested on several benchmarks, compared to a series of active learning baselines. The experimental results show interesting gains by the proposed method.

- The discussion on related work is carefully developed.

**Weaknesses:**

- Arguments and results on the effectiveness of the lexicographic order during query over the least-calibrated only strategy might lack consistency (see Questions).

- There might be some issues in the proofs (see Questions)

- Some notations are ambiguous or erroneous. According to the setup in the beginning of Section 2, $x_1$ stand for the first data vector in the initial labeled and unlabeled set at the same time. In Lines 221-222, the index $i$ goes from $0$ to $k$, querying $k+1$ samples instead of $k$, and creating repeated indices for samples queried at different rounds. The distribution of the cumulative labeled dataset $S$ is different from the initial label set $S_0$, meaning that we cannot have $S\sim\mathbb{P}(X)\mathbb{P}(Y\vert X)=\mathbb{P}(X,Y)$ and also $S_0\sim\mathbb{P}(X,Y)$, in contrary to the formula in (14). The abbreviation ECE for Expected Calibration Error is used before the specification of its meaning in Line 311.

**Questions:**

I have several questions regarding the lexicographic order for query and the proof of theoretical guarantees. First on the lexicographic order:

- At each query round, the proposed query strategy select first the least calibrated samples, then the least confident ones with zero calibration error. Does it mean that there is a Dirac at zero in the distribution of calibrated errors? If so, how to understand it?

- As explained by the authors, their method focuses on least-calibrated samples at early query rounds, then moves to the least confident ones later when the trained model is well calibrated. To illustrate the effectiveness of this lexicographic order, the authors compared the performance of their query strategy with querying by least-calibrated only in Figure 12, where the performance of least-calibrated only diverges already from the proposed method at very early rounds. Could the authors explain this observation while the two methods should be nearly identical at early query rounds?

On the proof of theoretical guarantees:

- Could the authors explain why the result of Popordanoska et al. (2022) can be applied to obtain (20), despite that the sample in the cumulative labeled set are not i.i.d. ?

- Could the authors explain in detail the passage from (42) to (43)? From where I see it, the fact that the average calibration error is smaller than $\epsilon$ on the samples queried at the $t$-th round does not imply that it is also the case for the samples queried at earlier rounds.

---

> ### Author Response · Authors · 2025-11-19
>
> Thank you for acknowledging our motivation for improving calibration in AL, our method’s idea, experimental evidence, and the discussion of related works. We updated our revision (news is colored in blue), and we address your concerns as follows:
>
> **W3. Notations.** We’ve fixed all notations, typos, and abbreviations in our revision. Regarding Eq.14, we fixed the marginal distribution $\mathbb{P}(X)$ to $\tilde{\mathbb{P}}(X)$ to denote the marginal distribution of samples selected from the model $h$. We use Eq.14 to prove from Eq.15 to Eq.19. Please note that these notations do not change any of our theoretical results.
>
> **W1-Q1. At each query round, the proposed query strategy select first the least calibrated samples, then the least confident ones with zero calibration error. Does it mean that there is a Dirac at zero in the distribution of calibrated errors?** No. Our query strategy in Eq.11-12. does not “select first the least calibrated samples, then the least confident ones with zero calibration error”. It sorts samples with the highest calibration errors in the unlabel pool. If samples share the same calibration error (not necessarily zero), then, within that group, it will continue to sort samples with the highest model uncertainty. Based on this sorted array, we will select the top-k samples. More details in numpy.lexsort (lexicographical sorting).
>
> **W1-Q2. In Fig.12, why are the proposed methods and the least-calibrated-only not nearly identical at early query rounds?** This is because in very early query rounds, many samples have the same calibration error in the unlabeled pool. Since this sample number exceeds the labeling cost, the least-calibrated-only method will randomly select from these same calibration error samples. In contrast, our proposed method, which utilizes lexicographical order, will continue sorting these samples by uncertainty to select them.
>
> **W2-Q3. Why can the result of Popordanoska et al. (2022) be applied to obtain Eq.(20)?** $n_t$ samples in the cumulative labeled set are not i.i.d. w.r.t. $\mathbb{P}(X,Y)$, but are i.i.d. w.r.t. $\mathbb{P}_s(X,Y) = \tilde{\mathbb{P}}(X)P(Y|X)$. Therefore, we can apply this result to our R.H.S. in Eq.20, which corresponds to the probability density $p_s$ for $\mathbb{P}_s(X,Y)$. Eqs. 20-21 results show that the estimator in Eq.9 is a point-wise consistent estimator under AL with covariate shift. This is also mentioned in Popordanoska’s work on general covariate shift [1]. We also added more explanations when deriving Eq.20 in our revision.
>
> [1] Teodora Popordanoska, et al. To trust or not to trust: Assessing calibration error under covariate shift without labels, 2023.
>
> **W2-Q4. Could the authors explain in detail the passage from Eq.(42) to Eq.(43)?** In Theorem 4.2., we state that “For every round $t\in (0,T]$, the trained model is assumed to have $\frac{1}{k_t}\sum_{j\in n_{t-1}+[k_t]}\left\|\mathbb{E}\left[Y|h(x_j)\right] -h(x_j)\right\|\_p^p \leq \epsilon$ over queried training points”. Therefore, we can bound Eq.(42) by Eq.(43) due to: $\sum_{i=1}^t\sum_{j=n_{t-1}+1}^{n_{t-1}+k_t} \left\|\mathbb{E}\left[Y|h(x_j)\right] -h(x_j)\right\|\_p^p \leq \epsilon\sum_{i=1}^t k_t$.

---

> ### Comment · Reviewer_yiPZ · 2025-11-25
>
> I thank the authors for their response. Please find below my follow-up comments.
>
> W1-Q1. The fact that many unlabelled data points share the same calibration error, whether it is zero or not, still suggests the presence of Diracs in the distribution of calibration error. As calibration error takes continuous value, it is still unclear to me why there are Diracs in its distribution .
>
> W1-Q2. Thanks for the clarification.
>
> W2-Q3. The samples in the cumulative labeled set are not i.i.d. w.r.t any distribution. Because the samples queried at the $t$-th round depend on all those queried at the $t-1$ earlier rounds, which breaks the independence.
>
> W2-Q4. If the authors meant to say that there is a universal value $\epsilon$ such that $\frac{1}{k_t}\sum_{j\in n_{t-1}+[k_t]}\left|\mathbb{E}\left[Y|h(x_j)\right] -h(x_j)\right|_p^p \leq \epsilon$ holds for all $t\in[1,T]$, then $\epsilon$ would be greater than the average calibration error at the first round, which is typically a large value. In that sense, Theorem 4.2 would hardly provide a control on the calibration error on unlabelled and test data.

---

> ### Author Response · Authors · 2025-12-03
>
> We thank the reviewer for your follow-up comments.
>
> **W1-Q1. Why are there Diracs in its distribution?** From our Eq.9, we can see that the calibration error is estimated using a specific bandwidth kernel parameter, and the calibration error is bounded between $[0,1]$. Therefore, in the early AL round, when the model is uncalibrated, there can be many samples with a calibration error of approximately 1 (Dirac at 1). In the later AL round, when the model is more calibrated, the calibration error across samples is reduced to 0 (as shown in Fig.3), which means that the calibration error histogram will be more concentrated around 0 (Dirac at 0).
>
> **W2-Q3. Samples in the cumulative labeled set are not i.i.d. w.r.t any distribution.** We agree that $n_t$ cumulative samples in the labeled sets are i.i.d. is a strong assumption in this AL setting. That said, for each round $t$, the queried samples $\\{(x_i,y_i)\\}\_{i={n_{t-1}+1}}^{n_t}$ can be i.i.d. as long as we have a probabilistic acquisition function for model $h(x)$. This is also discussed and mentioned in the covariate shift in AL [1,2]. Therefore, we can still apply Popordanoska et al. (2022)’s result on $n_0$ warm-up samples and $\\{(x_i,y_i)\\}\_{i={n_{t-1}+1}}^{n_t}$ samples. In the experiment, since the number of these samples is small, we used all $n_t$ samples to estimate the calibration error. Thanks for your comment. We explained this assumption in more detail in our revision.
>
> [1] Liu et al., Shift-Pessimistic Active Learning Using Robust Bias-Aware Prediction, AAAI, 2015.
>
> [2] Fannjiang et al., Conformal prediction under feedback covariate shift for biomolecular design, NAS, 2022.
>
> **W2-Q4**. Calibration error $\epsilon$ assumption. Regarding a small calibration error $\epsilon$ in the $k_t$ queried samples, we believe this is a mild assumption, as we can train these $k_t$ queried samples to minimize cross-entropy loss for every AL round.

---

### Author Response · Authors · 2025-12-03
**Final general response**

We thank all reviewers for reviewing our papers. We appreciate your acknowledgement of our motivation for improving calibration and accuracy in AL, our method’s idea, theoretical contributions on calibration error, experimental evidence, and the discussion of related works. We have responded to all reviewer comments to clarify the proposed problem of uncalibrated uncertainty sampling in AL, our method’s motivation, results, and further elaborated on our contributions. We also implemented the suggestion to consider comparison with using *Temperature Scaling on $100\\%$ queried training data (i.e., without hold-out recalibration set) of Reviewer cCs3 (despite it is not a standard practice for model calibration)*. Accordingly, we have updated our revision. Our main updates to the paper are summarized as follows:

- **Motivation (R-cCs3)**: We’ve added more explanation of the problem of uncalibrated uncertainty sampling in AL and our method’s motivation in the introduction. Specifically, the uncertainty sampling method in AL follows the intuition that the Acquisition Function (AF) should be “querying the most uncertain samples in the unlabeled pool because they may be the most inaccurate”. However, if the model is uncalibrated, then the most uncertain samples may not be the most inaccurate, resulting in inefficient uncertainty sampling AF. To address this issue, we propose our method to query the data points to resolve the uncalibrated problem. Once the model is calibrated (i.e., most uncertain samples are the most inaccurate), we can use “Calibrated Uncertainty Sampling” to query samples from the unlabelled pool to improve both the accuracy and calibration on the test set.

- **Comparison with post-hoc recalibration methods (R-cCs3)**: We’ve added Section 5.4 and Fig.7 to provide a more detailed comparison with recalibration methods. We also added the experiment of using TS on both hold-out recalibration and $100\\%$ cumulative dataset as requested by R-cCs3. From these results, we observe that our method outperforms all recalibration baselines in ECE and accuracy, confirming the benefits of our AF for AL.

- **Ablation study with other kernels (R-cCs3, R-1gMt)**: We’ve added Fig.19 to compare our default kernel (Dirichlet) with RBF-Gaussian, Cosine, and Linear. We observe that the performance differences across kernels are relatively small. That said, Dirichlet and RBF-Gaussian are still slightly better than Cosine and Linear with lower calibration and higher accuracy.

- **Notations and derivations (R-yiPZ)**: We corrected all notations (indexing, labeled distribution), typos, and abbreviations in our revision. We also added more discussion and reference about the i.i.d. assumption in the queried data in Section 4 and our proof.

- **Evaluation of additional Kolmogorov-Smirnov (KS) calibration error metric (R-1gMt)**: We’ve added Fig.18 to confirm that our lowest calibration error result is consistent across bining-free KS and bining-based ECE calibration metrics.

Overall, we thank the reviewers for their feedback and appreciate the opportunity to further improve our paper.

---

### Meta-Review · Area_Chair_WDE6 · 2025-12-22

**Summary:**

The paper proposes an active learning acquisition function that prioritizes querying samples with high estimated calibration error, motivated by the observation that uncertainty-based querying can be unreliable when model uncertainty is poorly calibrated. Using a kernel-based calibration error estimator under covariate shift, the paper provides theoretical guarantees of bounded calibration error on both the unlabeled pool and test data, and report empirical gains in calibration and generalization over standard acquisition baselines.

Reviewers raised several critical concerns, including the lack of convincing justification for the proposed lexicographic ordering, limited guidance on kernel choice, unclear motivation of the acquisition function, missing comparisons with post-hoc calibration methods, and potentially unrealistic assumptions underlying the theory. They also noted issues in the experimental results and possible errors in the proof.

While the rebuttal clarifies some points, key concerns remain. In particular, as one reviewer observed, the unbiased estimator appears to rely on having a sufficiently large labeled set, an assumption that does not hold with the label-scarce regime inherent to active learning. In fact, reliably estimating calibration error in a label-limited setting is highly non-trivial, which may also help explain the empirical issues highlighted by reviewers. The authors are encouraged to incorporate these constructive comments to strengthen the work for a future submission.

**Reviewer Concerns:**

The rebuttal clarifies several points, including the kernel choice, the rationale for the lexicographic ordering, and comparisons with post-hoc calibration methods. However, major concerns remain unresolved, particularly the empirical issues, likely stemming from unreliable calibration error estimation in a label-scarce active learning setting, and the unrealistic assumptions underpinning the theoretical guarantees.

**Reviewer Scores:**

Given that the key concerns around the reliability of calibration-error estimation and the realism of the theoretical assumptions remain unresolved, reviewers’ scores are likely to remain unchanged.

---

### Decision · Program_Chairs · 2026-01-26

Reject